# Modifying muscle metabolic dysregulation in inclusion body myositis with pioglitazone: a single-arm trial

Brittany L. Adler [1,9] ✉, Michael R. Bene [2,9], Cissy Zhang[3], Conrad Say[4], Pratik Khare[3,4], Christopher Mecoli[1], Eleni Tiniakou[1], Julie J. Paik[1], Tae Chung[5], Erika Darrah [1], Thomas Lloyd[6], Ruben Pagkatipunan[5], Megan McGowan[5], Albert Mears[5], Hilary J. Vernon[7], Lisa Christopher-Stine [1,6], Anne Le[3,4,8] & Jemima Albayda [1] ✉

This single-arm, open-label phase 1 trial evaluated the PPARγ agonist pioglitazone in patients with inclusion body myositis (IBM). After a 16-week observation (lead-in) period, participants received pioglitazone 45 mg daily for 32 weeks. The primary outcome was the change in PPARGC1A expression and related metabolic pathways in muscle after 16 weeks of treatment compared with the lead-in period. Of the 16 enrolled participants, 13 initiated pioglitazone and completed at least one on-treatment assessment; the trial was terminated early due to the COVID-19 pandemic. At baseline, muscle metabolomics revealed broad metabolic abnormalities compared with controls. Pioglitazone reversed elements of this signature, increasing PPARGC1A expression ($p = 0.099$) and modulating downstream pathways in muscle, including enhanced oxidative phosphorylation. Clinical outcomes were unchanged overall, but a subset with favorable metabolic responses showed slower decline in the IBM-Functional Rating Score (IBM-FRS) and Modified Timed Up and Go (m-TUG). Reported adverse effects included myalgia and heart failure exacerbation. As a phase 1 trial with a limited cohort, these findings provide preliminary evidence that pioglitazone modulates muscle metabolism and warrants further investigation in IBM. This study was supported by the Ira T. Discovery Fund and the Peter and Carmen Lucia Buck Foundation Myositis Discovery Fund. Clinical Trials Registration: NCT03440034.

Sporadic inclusion body myositis (IBM) is the most common inflammatory myopathy in the elderly[1]. It differs from other types of myositis by its insidious onset with slowly progressive weakness primarily affecting the quadriceps and finger flexors, and a lack of significant response to immunosuppression. Although the clinical course is highly variable, IBM leads to significant disability with the median time from onset of symptoms to wheelchair use of about 10 years[2]. A barrier to finding effective treatment is that its pathophysiology remains

[1]Division of Rheumatology, Department of Medicine, Johns Hopkins School of Medicine, Baltimore, MD, USA. [2]Division of Geriatric Medicine and Gerontology, Johns Hopkins School of Medicine, Baltimore, MD, USA. [3]Gigantest, Inc, 31 Light Street, Baltimore, MD, USA. [4]Department of Biology, Johns Hopkins University, Krieger School of Arts and Sciences, Baltimore, MD, USA. [5]Department of Physical Medicine and Rehabilitation, Johns Hopkins School of Medicine, Baltimore, MD, USA. [6]Department of Neurology, Johns Hopkins School of Medicine, Baltimore, MD, USA. [7]Department of Genetic Medicine, Johns Hopkins School of medicine, Baltimore, MD, USA. [8]Department of Pathology and Oncology, Johns Hopkins School of Medicine, Baltimore, MD, USA. [9]These authors contributed equally: Brittany L. Adler, Michael R. Bene. ✉e-mail: brit.adler@jhmi.edu; jalbayd1@jhmi.edu

incompletely understood. It is known to share features of both inflammation (infiltration of CD8 + T cells, MHC class I/II, pro-inflammatory cytokine upregulation, association with the anti-cN1a autoantibody) as well as degeneration (presence of rimmed vacuoles, sarcoplasmic aggregation of proteins, mitochondrial dysfunction)[3]. Multiple pharmacologic treatments targeting both the immune system as well as various degenerative aspects of the disease have failed to show clinical benefit[4], and there is a need to explore additional mechanisms and novel targets.

One of the key characteristics of IBM is the presence of abnormalities in skeletal muscle mitochondria, which are hypothesized to play a role in the pathophysiology of this disease. Muscle biopsies from patients with IBM often display ragged red fibers, deficiency in activity of the electron transport protein cytochrome C oxidase, abnormal proliferation and subsarcolemmal accumulation of mitochondria, and large-scale mitochondrial deoxyribonucleic acid deletions[5–7]. In addition, endurance exercise, which activates key mitochondrial regulators and stimulates mitochondrial biogenesis, improves muscle strength and slows disease progression in IBM[8,9]. AMP-Activated Protein Kinase (AMPK) and Peroxisome Proliferator-Activated Receptor gamma coactivator 1-alpha (PGC-1α) are key regulators of mitochondrial function that are activated by endurance exercise[10,11]. In a rat model of IBM, PGC-1α expression was significantly reduced as compared to normal rats, and expression of PGC-1α and severity of the IBM phenotype were rescued by resistance exercise[12]. This raises the possibility that mitochondrial dysregulation and the upstream molecular pathways that impact mitochondrial function and metabolism may be promising therapeutic targets for stabilizing the disease and improving strength in IBM.

Pioglitazone, a thiazolidinedione medication used to treat type 2 diabetes, targets the nuclear receptor peroxisome proliferator-activated receptor gamma (PPARγ) and increases expression of AMPK and PGC-1α[13]. Pioglitazone enhances mitochondrial function, exercise capacity, and mitochondrial biogenesis in skeletal muscle[13–17], which prompted evaluation of whether it could modulate similar pathways in IBM. We conducted a proof-of-concept, open-label clinical trial of pioglitazone in IBM to test this hypothesis. Here, we report that pioglitazone treatment was associated with observable transcriptomic and metabolic changes in IBM muscle. These findings suggest targeting mitochondrial pathways warrants further investigation in IBM.

## Results

We utilized a single-arm study design in which all patients were observed off therapy for 16 weeks to assess the trajectory of their disease (lead-in period), and then crossed over to receive a total of 32 weeks of pioglitazone therapy (intervention period). Muscle and serum samples were obtained along with functional measures at the baseline visit, week 16 (end of the lead-in period), and week 32 (after 16 weeks of pioglitazone treatment). At week 48, only serum and functional measures were obtained (see study schema, Fig. 1).

### Study participants

Among the 20 patients who were screened for eligibility, 16 patients met the inclusion criteria and were enrolled. Of the 16 enrolled participants, 3 did not initiate pioglitazone. One participant withdrew due to a fracture sustained prior to treatment initiation, one withdrew due to concerns related to the COVID-19 pandemic, and one was withdrawn due to noncompliance. As a result, 13 participants initiated pioglitazone therapy and were included in the on-treatment analysis (Fig. 1).

The mean age of the enrolled IBM participants was $66.8 \pm 7.3$ years, 50% were women, and 100% were white. The average duration of IBM since symptom onset was $5.4 \pm 4.4$ years and 4/16 (25%) used an assistive device. The mean IBM-FRS at baseline was $28.9 \pm 3.9$ and creatine kinase (CK) was $498 \pm 384$ U/L. The mean age of the 5 control muscle specimens was $60.6 \pm 15.4$ years and 4/5 (80%) were female. The 10 control sera were slightly younger than the IBM sera ($56.3 \pm 5.7$ years, $p = 0.001$), and 7/10 (70%) were female. Supplementary Tables 1 and 2 include additional demographic and clinical features of the IBM patients and controls.

### IBM muscle exhibits a distinct metabolic signature

To determine the effect of pioglitazone on the muscle metabolome in IBM, we first identified the metabolic signature of IBM by comparing metabolites in muscle from IBM patients at study baseline with that of healthy controls. Of the 16 IBM study patients who underwent a baseline muscle biopsy, one sample was excluded due to poor tissue quality. Untargeted metabolomics was performed on the 15 remaining samples and compared to five healthy control muscle specimens obtained from the Johns Hopkins Neuromuscular Biobank. Using principal component analysis, we found that samples from healthy control and IBM muscle showed distinct clustering patterns, with a more variable clustering of IBM samples than healthy controls (F-Test on PC1, $p = 0.018$) (Fig. 2A). Several key metabolites involved in glycolysis, pyrimidine metabolism, tryptophan metabolism, and oxidation-reduction (redox) reactions were significantly different in IBM muscle compared to healthy controls (Fig. 2B and Supplementary Table 3). Pathway analysis using KEGG Pathways

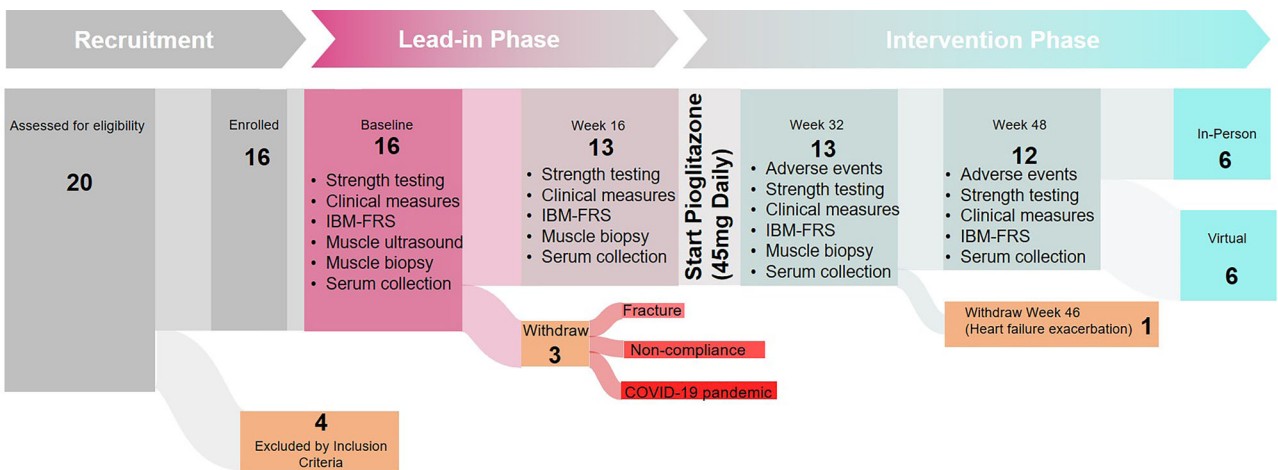

**Fig. 1 | Study design and timeline.** Patients underwent a 16-week lead-in phase without therapy, followed by 32 weeks of pioglitazone treatment. Of 16 enrolled participants, 3 did not initiate pioglitazone (fracture, COVID-19 pandemic-related withdrawal, noncompliance). Clinical outcome measures were assessed longitudinally, and muscle biopsies were performed at baseline, week 16, and week 32. IBM-FRS Inclusion Body Myositis Functional Rating Scale.

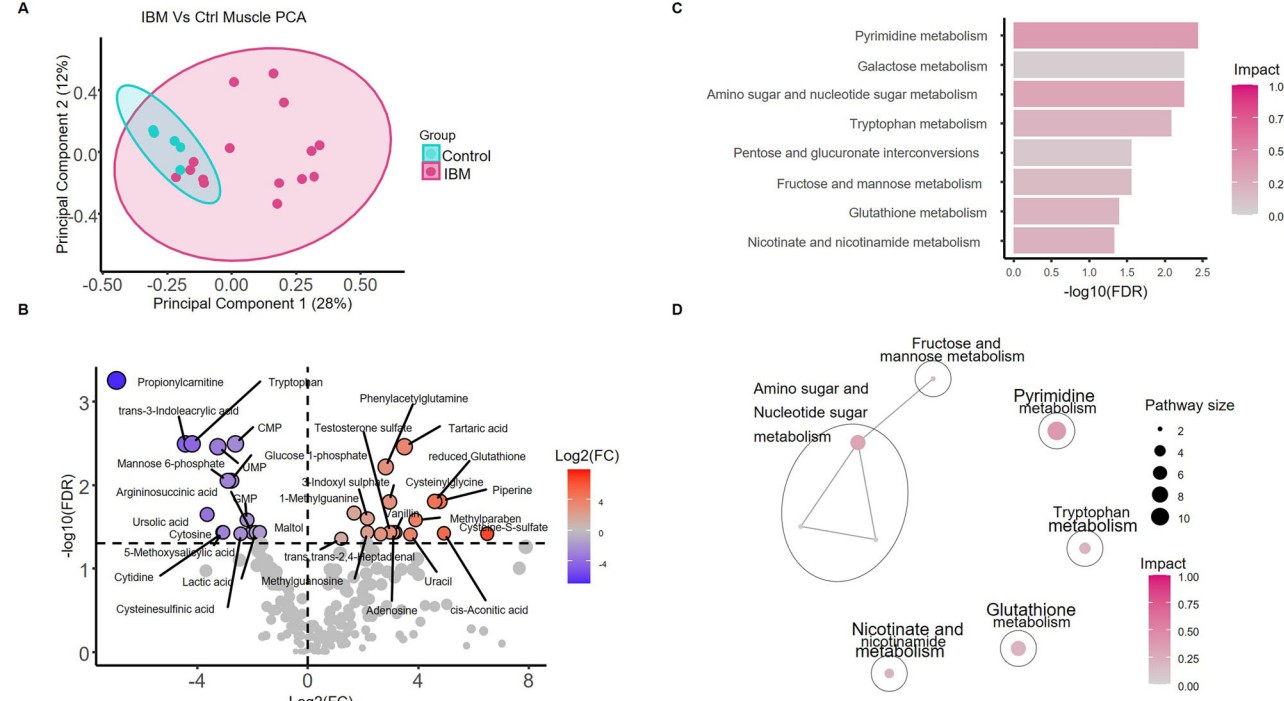

**Fig. 2 | Untargeted metabolomics analysis of baseline muscle samples from inclusion body myositis (IBM, *n* = 15) and controls (*n* = 5). A** Principal component analysis (PCA) of all detected metabolites in muscle tissue. Cyan control samples, fuchsia IBM **B** Volcano plot illustrating the comparative abundance of metabolites between IBM and control muscle, where FC fold change. Red = elevated in IBM compared to control, Blue = reduced in IBM compared to control samples. **C** KEGG pathway analysis highlighting the most significant differentially abundant

metabolic pathways between IBM and controls (FDR < 0.05). Color intensity reflects the pathway impact score. **D** Network diagram of clustered pathways, with each node representing a significant pathway, edges indicating shared metabolites, black circle indicate pathway clusters, and representative pathways for clusters labeled. Source data are provided as a Source data file. KEGG Kyoto Encyclopedia of Genes and Genomes.

identified 8 pathways that were differentially expressed in IBM vs controls, which included pyrimidine metabolism, amino acid and nucleotide sugar metabolism, tryptophan metabolism, and glutathione metabolism (Fig. 2C, D).

Given prior implications of disturbed mitochondrial function and bioenergetics in IBM, we next specifically analyzed metabolites associated with glycolysis, the TCA cycle, and oxidative stress. We first focused on glycolysis and the TCA cycle in IBM muscle tissue and identified significant abnormalities in key metabolites, including lactate and glucose-1-phosphate, which were significantly decreased in IBM muscle (Fig. 3B, lactate: log2FC −1.89, FDR = 0.037; glucose-1-phosphate: log2FC −2.78, FDR = 0.009). In the TCA cycle, the TCA intermediate cis-aconitate was significantly elevated (log2FC 4.93, FDR = 0.038), with citric acid and α-ketoglutarate also trending higher in IBM muscle. In contrast, distal TCA metabolites such as succinate, fumarate, and malate trended lower in IBM muscle (Fig. 3B, Supplementary Table 4). Propionylcarnitine, a metabolite in equilibrium with propionyl-CoA which feeds into the TCA cycle through conversion to succinyl-CoA, was significantly reduced (log2FC −6.93, FDR = 0.0006) in IBM (Fig. 2B), and may exacerbate distal TCA cycle dysfunction.

We next examined additional metabolites that contribute to replenishing intermediates of the TCA cycle, including pyrimidines[18]. Key pyrimidine metabolites, including the nucleoside cytidine (log2FC −3.06, FDR = 0.037) and the nucleotides cytosine, CMP, UMP, and GMP, were among the most reduced metabolites in IBM (Fig. 2B, Supplementary Table 3). As for nucleotide metabolism, while most purine and pyrimidine metabolites were significantly decreased in IBM muscle, we observed increased uracil (log2FC 3.71, FDR = 0.039) and adenosine (log2FC 3.09, FDR = 0.037), as well as methylated nucleotides such as 1-methylguanine (log2FC 1.68, FDR = 0.022) and methyl-guanosine (log2FC 2.15, FDR = 0.037)[19].

Given abnormalities in mitochondrial function, we examined differences in redox homeostasis. Compounds such as 3-indoxyl sulfate, which is known to promote the production of reactive oxygen species (ROS) and thus disrupt redox homeostasis[20], were increased in IBM muscle (log2FC 2.16, FDR = 0.025), whereas metabolites that alleviate oxidative stress, such as indole-3 acry-late, were decreased (log2FC −4.44, FDR = 0.003). Overall, redox homeostasis was disrupted, as reflected by increased intensities of metabolites that induce reactive oxygen species, and decreased metabolites that moderate oxidative stress (Fig. 3A, Supplementary Tables 3 and 4).

**Metabolic dysregulation in IBM muscle and disease severity**

We next sought to determine if metabolic dysregulation is associated with disease severity in IBM. The IBM patients in our cohort exhibited varying degrees of muscle involvement, and we stratified muscle disease severity by muscle ultrasound (US) to capture a muscle quality/structural assessment. We evaluated for increased echointensity and atrophy in the flexor digitorum profundus, rectus femoris, and vastus lateralis muscles, which indicates chronicity and damage. Patients with a US sum score >10 were characterized as having severe disease (*n* = 7), and patients with a US sum score ≤10 were characterized as having mild-moderate disease (*n* = 6)[21,22] (Fig. 4A, B). This stratification correlated significantly with function, and those with severe disease by US had lower IBM-FRS scores at baseline (Spearman's *p* = −0.62, *p* = 0.03) (Supplementary Table 5). We performed a correlation analysis of the log2FC values of 211 muscle metabolites comparing the fold change between IBM vs. controls (*x*-axis) and severe vs mild-moderate IBM (*y*-axis). There was a positive moderate correlation (Spearman's *p* = 0.44, *p* < 0.0001), suggesting that the metabolic signature of IBM in

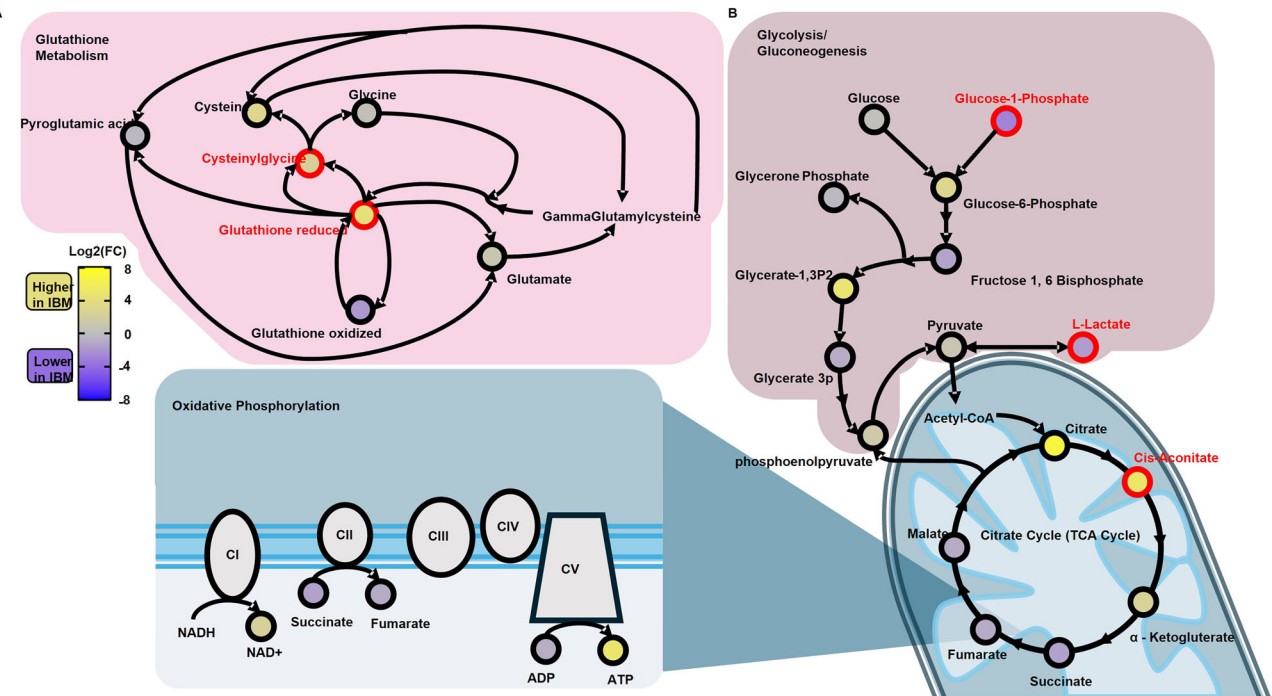

**Fig. 3 | Metabolic pathways in IBM skeletal muscle.** Detected metabolites in KEGG pathways for **A** glutathione metabolism and **B** glycolysis, TCA cycle, and oxidative phosphorylation in muscle tissue from patients with inclusion body myositis (IBM, $n = 15$) compared to controls ($n = 5$). The color gradient represents the Log2 fold change (log2FC), with yellow representing metabolites with greater abundance in IBM and blue those with lower abundance in IBM samples. Metabolites highlighted in red indicate a false discovery rate (FDR) < 0.05. Source data are provided as a Source data file. KEGG Kyoto Encyclopedia of Genes and Genomes. TCA tricarboxylic acid. CI – CV = complex I – V. NADH nicotinamide adenine dinucleotide plus hydrogen. NAD + = Nicotinamide adenine dinucleotide. ADP Adenosine diphosphate. ATP Adenosine triphosphate.

muscle is more accentuated in severe IBM (Fig. 4C). We did not observe a statistically significant difference in the metabolic signature of severe versus mild-moderate IBM in sera (Spearman's $p = -0.13$, $p = 0.06$) (Fig. 4D).

### Metabolic signature in IBM muscle does not mirror serum

After identifying multiple differences in the metabolic signatures of affected IBM muscle versus normal control muscle, we examined differences between the serum metabolic profile in patients with IBM and healthy controls to determine if differences seen in muscle metabolism were reflected in serum. As observed with muscle, the metabolic profile of serum from patients with IBM exhibited distinct clustering compared to healthy controls (Fig. 5A). Specifically, 25 metabolites (Fig. 5B, Supplementary Table 6) and 25 pathways (Fig. 5C) were differentially expressed in IBM serum relative to that from healthy controls. Metabolites that were decreased in IBM serum included carnosine (log2FC −5.99, FDR = 4.54E-14), propionylcarnitine (log2FC −5.66, FDR = 1.13E-07), indole-3-acrylic acid (log2FC −7.11, FDR = 1.10E-07), and tryptophan (log2FC −5.72, FDR = 1.13E-07). Pathway enrichment analysis identified differentially expressed pathways involving beta-alanine metabolism, histidine, tryptophan metabolism, and pentose phosphate pathways, among others (Fig. 5C, D).

To compare the global metabolic signatures of IBM between serum and muscle, we performed a correlation analysis using the Log2 fold change (log2FC) values of the 124 shared metabolites. Despite notable agreement for a few key metabolites, such as propionylcarnitine and tryptophan which were both decreased in serum and muscle, the overall correlation between the metabolic profiles in muscle and serum was not significant (Fig. 5E, Spearman's $p = 0.06$, $p = 0.5$). These findings suggest that IBM has distinct effects on metabolites in serum and muscle.

### Pioglitazone induces favorable metabolic changes in IBM muscle

Of the 16 enrolled participants with IBM, 13 received pioglitazone therapy and underwent at least one on-treatment clinical evaluation and muscle biopsy, following a 16-week lead-in period. We performed RNA sequencing and untargeted metabolomics to identify transcriptomic and metabolomic changes with pioglitazone for each patient compared to the 16-week lead-in period.

**i. Primary outcomes.** The primary outcome was the change in expression of *PPARGC1A* and other oxidative phosphorylation and mitochondrial pathway genes after treatment with pioglitazone compared to the lead-in period. We did not identify a significant change in *PPARGC1A* gene expression during the treatment-phase compared to the lead-in phase using a paired t-test ($p = 0.099$, mean difference = 1.19[−0.26,2.64]), although there was a trend toward increased expression after treatment. Notably, *PPARCC1B* gene expression, which is not a known target of pioglitazone and thus served as a control, did not change with pioglitazone (Fig. 6A). As PGC-1α is a transcriptional coactivator, changes in its activity may be better assessed through identification of alterations in its target gene expression. We therefore utilized Ingenuity Pathway Analysis to assess the change in *PPARGC1A* regulated gene expression, which identified *PPARGC1A* as a significantly activated upstream regulator in the treatment phase (Activation z-score = 5.183, padj. = 0.00014). In contrast, *PPARGC1A* was non-significantly inhibited in the lead-in phase (Activation z-score = −3.028, padj. = 0.123).

Gene set enrichment analysis of KEGG pathways identified 56 differentially expressed pathways during pioglitazone treatment (Supplementary Table 7B), compared to 16 differentially expressed pathways during the lead-in period (Supplementary Table 7A). After 16 weeks of pioglitazone, genes involved in the TCA cycle, OXPHOS, PPAR

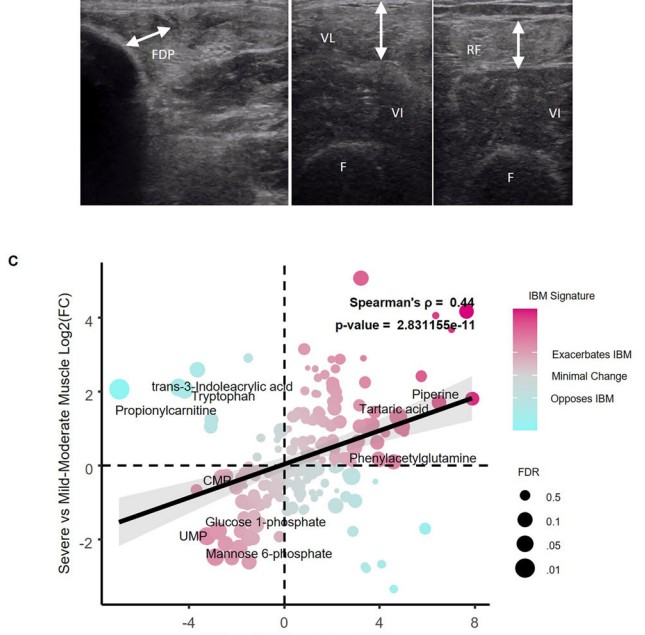

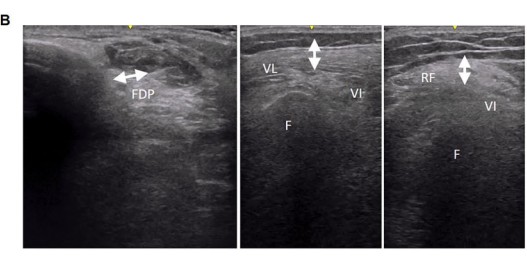

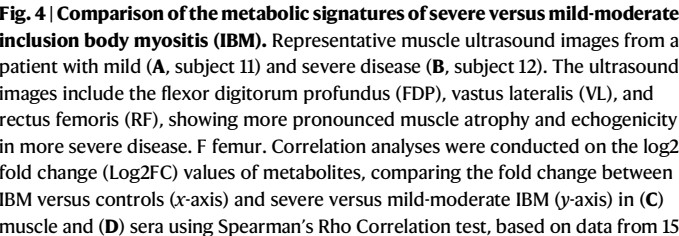

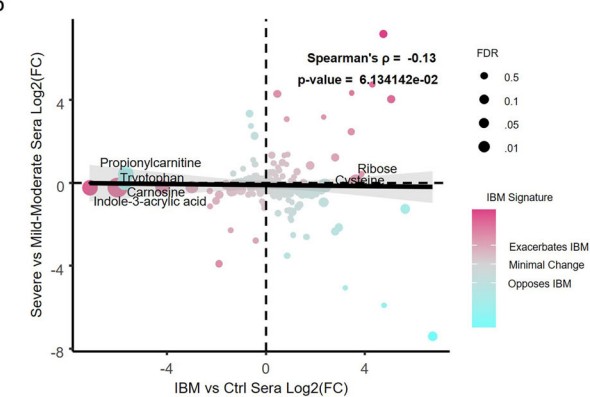

**Fig. 4 | Comparison of the metabolic signatures of severe versus mild-moderate inclusion body myositis (IBM).** Representative muscle ultrasound images from a patient with mild (**A**, subject 11) and severe disease (**B**, subject 12). The ultrasound images include the flexor digitorum profundus (FDP), vastus lateralis (VL), and rectus femoris (RF), showing more pronounced muscle atrophy and echogenicity in more severe disease. F femur. Correlation analyses were conducted on the log2 fold change (Log2FC) values of metabolites, comparing the fold change between IBM versus controls (x-axis) and severe versus mild-moderate IBM (y-axis) in (**C**) muscle and (**D**) sera using Spearman's Rho Correlation test, based on data from 15 IBM patients, 5 healthy muscle controls, and 10 healthy serum controls. Metabolites highlighted in fuchsia are shifted in advanced IBM in the direction of the IBM metabolic signature, while metabolites in cyan are shifted in advanced IBM in the direction of the controls. The significance of the correlation was determined using Spearman's correlation coefficient, with FDR representing Benjamini-Hochberg corrected Fisher's combined p-values across both comparisons (FDR < 0.05 labeled). Black line is a linear regression with 95% CI for visualization purposes. Source data are provided as a Source data file. UMP uridine monophosphate.

signaling, and AMPK signaling were significantly upregulated. In contrast, during the 16-week lead-in period, these same pathways, including OXPHOS and the TCA cycle, were downregulated (Fig. 6B). Critically, of the pathways significantly upregulated in the treatment phase were 8/10 *PPARGC1A* containing KEGG pathways. Two key pathway clusters—metabolism and inflammation/immune dysregulation—were identified. Before treatment, metabolic pathways were downregulated, and inflammatory pathways, such as IL-17 and cytokine interactions, were upregulated. During the treatment phase, an opposite pattern was observed in which metabolic pathways were upregulated and inflammatory activity appeared reduced, a pattern contrasting with that of the lead-in phase (Fig. 6B).

**ii. Secondary outcomes.** We next investigated the effect of pioglitazone on the overall metabolic signature of IBM, which was one of the secondary outcomes. One patient was excluded from these analyses because one of their longitudinal muscle samples was of poor quality. Initially, we compared the metabolic signature of IBM to the change in metabolites during the lead-in period (baseline to week 16) and observed no significant changes (Spearman's $p = 0.03$, $p = 0.65$), indicating the relative stability of the metabolic signature over the 16-week lead-in period (Fig. 7A). We then performed a similar analysis of the treatment period, comparing the IBM metabolic signature (x-axis) to the change in metabolites during treatment (y-axis). In contrast to the lead-in period, there was a measurable change in the metabolic profile over the 16 weeks of pioglitazone treatment. The negative correlation (Spearman's $p = -0.34$, $p < 0.0001$) indicates that pioglitazone was associated with modest changes in the muscle metabolome, with patterns that trended toward the profile of healthy controls and away from the metabolic signature of IBM (Fig. 7B).

We then performed an integrated pathway analysis using the transcriptomic and metabolomic results. These outcomes largely aligned with those observed in the transcriptomic pathway analysis (Fig. 6B), suggesting increasing inflammation during the lead-in phase, which was reduced in the treatment-phase, as well as a reversal of metabolic dysregulation in the treatment phase (Supplementary Table 8A, B). Finally, while we did not have sufficient non-diseased control samples for transcriptome analyses, we sought to evaluate if the observed effects on the IBM metabolic signature were recapitulated on another "omics" layer. We therefore utilized a publicly available data set with control and IBM muscle transcriptome samples to generate an external IBM transcriptomic signature with which to compare (GSE102138)[23]. Notably, a similar correlation analysis of our transcriptome data compared to this external IBM transcriptomic signature, detecting a significant shift in the transcriptome to becoming more "IBM-like" during the lead-in phase (Spearman's $p = 0.4$, $p < 0.0001$) and a significant shift to becoming less "IBM-like" during the treatment phase (Spearman's $p = -0.54$, $p < 0.0001$). This analysis provides valuable validation of our findings on another "omic" layer and with use of an external IBM cohort (Supplementary Fig. 1).

The effect of pioglitazone was heterogenous, with a positive metabolic response observed in only a subset of patients (4 out of 12, or 33% of patients) (Fig. 7C). In an exploratory analysis, metabolic responses varied by disease severity as stratified by US, with less affected individuals showing larger changes (treatment × severity, $p = 0.013$) (Figs. 4A, B and 7D), indicating that treatment-associated metabolic shifts were more evident in participants with greater muscle reserve. Two of the patients in the mild-moderate category had a strong metabolic response to pioglitazone, which resulted in a significant shift in their metabolic profile opposite that of the IBM

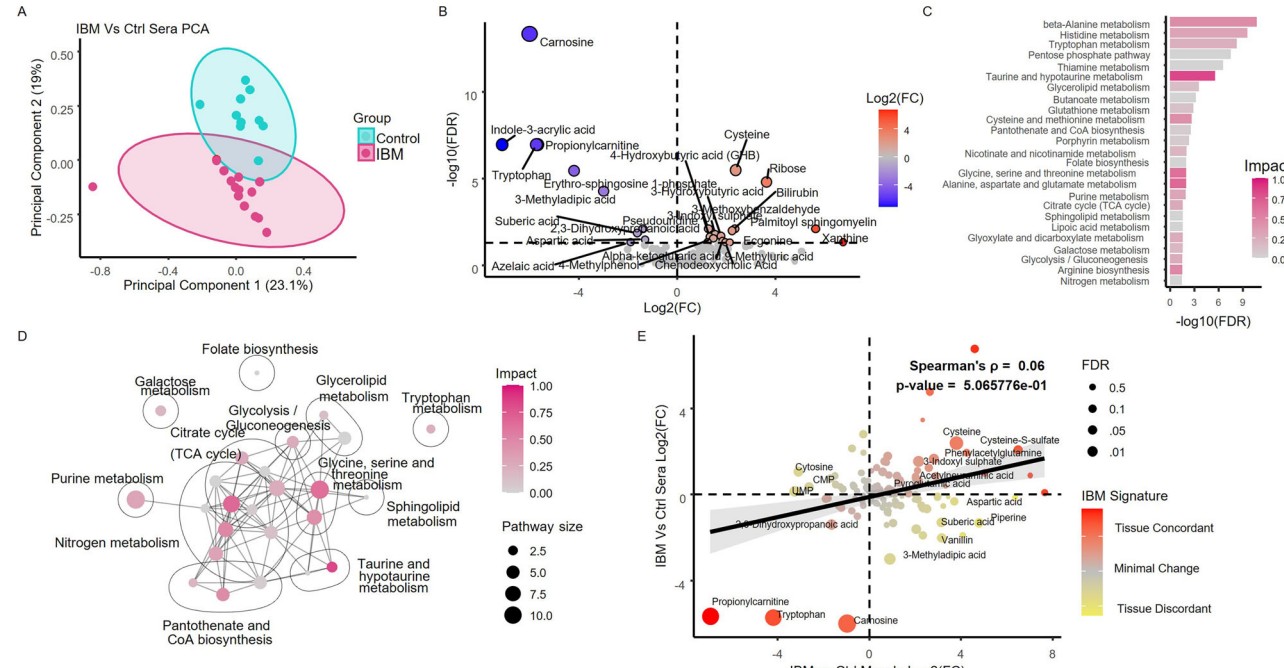

**Fig. 5 | Untargeted metabolomics analysis of baseline serum samples from inclusion body myositis (IBM, $n = 15$) and controls ($n = 10$). A** Principal component analysis (PCA) of all detected metabolites in serum. Cyan = control samples, fuchsia = IBM **B** Volcano plot illustrating the comparative abundance of metabolites between IBM and control serum. **C** KEGG pathway analysis identifying the most differentially abundant metabolic pathways in IBM versus control serum (FDR < 0.05). **D** Network diagram of clustered pathways, with each node representing a significant pathway, edges indicating shared metabolites, black circle indicate pathway clusters, and representative pathways for clusters labeled. **E** Correlation analysis of the fold change (FC) in metabolites between IBM and controls in muscle (*x*-axis) and serum (*y*-axis). Metabolites highlighted in red are shifted in the same direction in both muscle and serum relative to the IBM metabolic signature, whereas metabolites in yellow are shifted in opposite directions. The significance of the correlation was determined using Spearman's correlation coefficient, with FDR representing Benjamini–Hochberg corrected Fisher's combined *p*-values across both comparisons (FDR < 0.05 labeled). Black line is a linear regression with 95% CI for visualization purposes. Source data are provided as a Source data file. TCA tricarboxylic acid. UMP uridine monophosphate. CMP cytidine monophosphate.

metabolic signature (Fig. 7D) A sensitivity analysis, however, suggested the significant disease severity-metabolic response relationship was strongly influenced by one of these patients, with exclusion of them from the analysis resulting in a substantial increase in *p*-value for the interaction (from $p = 0.013$ to $p = 0.103$), necessitating caution in the interpretation of these results. Notable metabolites that increased with pioglitazone treatment in muscle included UMP, CMP, propionylcarnitine, tryptophan, and mannose-6-phosphate, all of which were significantly reduced in IBM compared to controls and were further reduced in severe compared to mild-moderate IBM. There was no significant correlation with pioglitazone treatment and metabolites in serum over the same period (Spearman's $p = -0.01$, $p = 0.84$) (Supplementary Fig. 2), indicating that the metabolic changes observed with pioglitazone were confined to muscle tissue.

## Safety and clinical outcomes with pioglitazone in IBM

The safety profile of pioglitazone in this study was similar to what has previously been reported in other disease settings[24,25]. There was no significant effect of treatment on weight (Treatment [Pio] × Time, 1.3 [−2.16–4.76], $p = 0.451$) or pro-BNP (Treatment [Pio] × Time, 117.78 [−24.35 −259.91], $p = 0.102$). Of the 13 patients with IBM who initiated pioglitazone therapy, 2/13 patients experienced an adverse event that was attributed to study therapy: one with myalgia and another with a heart failure exacerbation. Our patient did not have any clinical signs or symptoms of heart failure at the time of enrollment but had this in their remote history as occurring after a virus, which then resolved. Two weeks prior to termination of the trial, the patient felt a sense of chest discomfort, which gradually worsened with the finding of left-sided pleural effusion. Pioglitazone was stopped and diuretics were given with resolution of symptoms.

Aside from the single patient who discontinued study therapy due to an adverse event, all other patients (12/13, 92%) continued pioglitazone therapy until the final clinical evaluation at week 48, which was to assess the impact of extended use of the medication. However, 6 out of 12 final clinical evaluations were conducted virtually due to restrictions related to the COVID-19 pandemic. In-person functional measures, all of which were secondary outcome measures, included knee extensor dynamometry, m-TUG, 6-min walk test, hand-grip strength, IBM-FRS, serum CK level, composite Fi-2, and self-reported number of falls. For patients who were unable to present in person for the final visit due to the COVID-19 pandemic, only the IBM-FRS and number of falls were assessed virtually and serum CK levels were measured.

We next examined the impact of pioglitazone on the clinical secondary outcome measures. There was no significant change in any clinical outcome during the 16-week lead-in period ($p > 0.05$ for all comparisons) (Table 1). IBM is a slowly progressive disease, and these data suggest that the baseline rate of change in patients was below the level of detection of the clinical outcome measures. We subsequently assessed changes in clinical measures after administration of pioglitazone using a mixed effects model to determine the effect of pioglitazone on these same outcome measures. We did not observe a significant change in any outcome measure after administration of pioglitazone (Treatment[Pio] and Time × Treatment[Pio], $p > 0.05$ for all outcomes) (Table 1).

We next assessed the relationship between metabolic response in muscle and clinical outcomes in an exploratory analysis. We calculated individual metabolic correlation scores (rho) for each patient during the lead-in period (baseline to week 16) and during the intervention period (week 16 to week 32). A more positive rho (red)

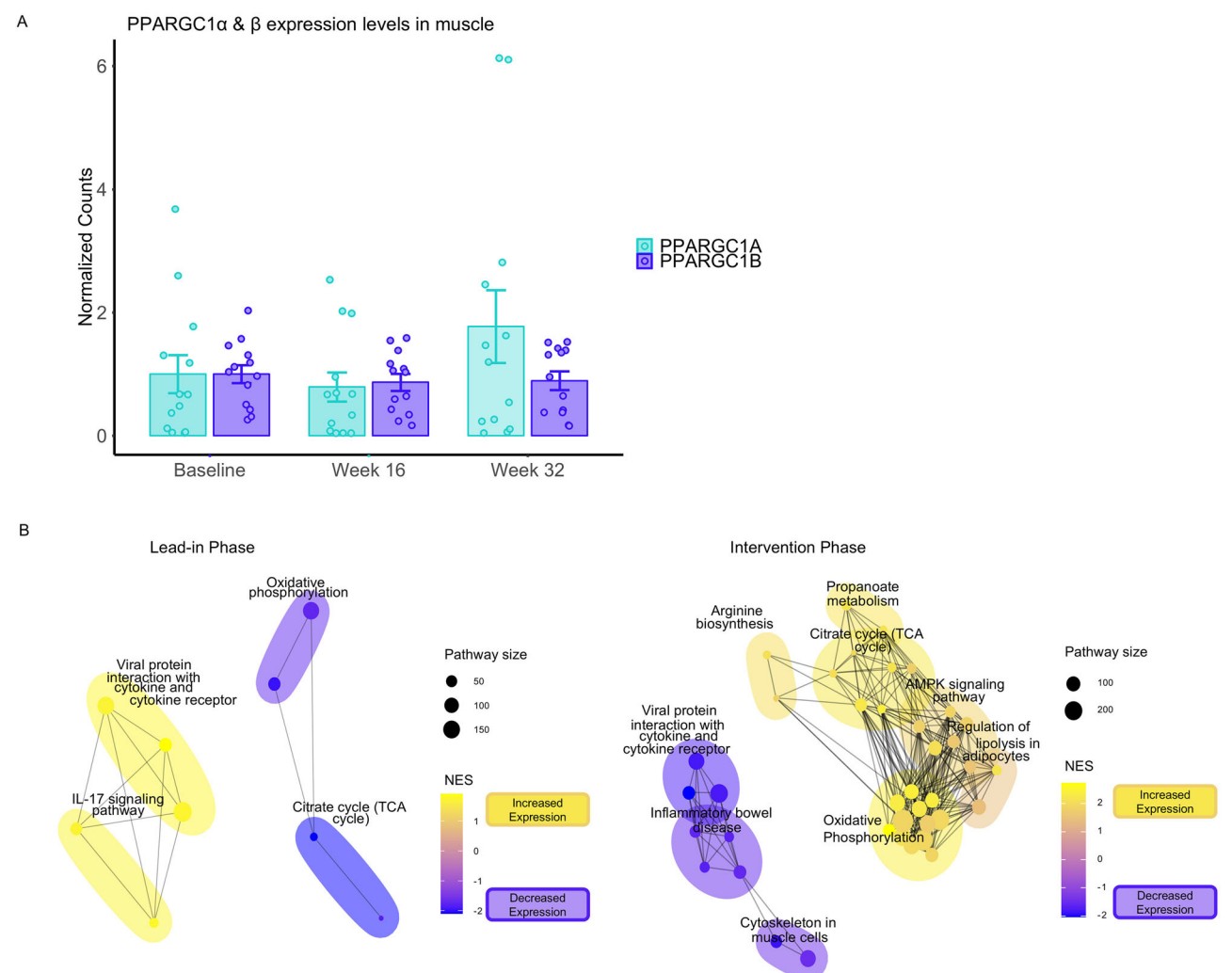

**Fig. 6 | Transcriptomic changes with pioglitazone in patients with inclusion body myositis (IBM). A** Mean gene expression levels of PPARGC1α at baseline, week 16 (end of the lead-in period), and week 32 (after 16 weeks of pioglitazone treatment) colored in cyan. PPARGC1β gene expression, which is not a known target of pioglitazone, is shown as a control colored in purple. Bars represent the mean ± SEM expression across 13 independent biological samples, dots represent individual data points. **B** Gene set enrichment analysis using KEGG pathways identified differentially expressed pathways during the lead-in period (baseline to week 16) and during pioglitazone treatment (week 16 to week 32). Network diagram of clustered pathways are displayed, with each node representing a significant pathway, edges indicating shared genes, and representative pathways labeled. Yellow are pathways with elevated expression and purple pathways with reduced expression. NES normalized enrichment score. PPARGC Peroxisome proliferator-activated receptor gamma coactivator. IL-17 Interleukin-17. TCA tricarboxylic acid. Source data are provided as a Source data file.

indicates that the patient's metabolite signature was shifted in the same direction as IBM, while a more negative rho (blue) indicates a shift that opposes the IBM metabolic signature. We generated a mixed-effect model to determine the effect of metabolic response (rho) on each clinical outcome measure. There was no significant rho x time effect for Fi-2, hand-grip strength, CK, 6-min walk, and knee extensor dynamometry outcome measures. However, there was a significant rho x time effect for both the IBM-FRS and the m-TUG, suggesting that a metabolic shift away from the IBM signature was associated with a less severe decline in the IBM-FRS over time and a slower increase in m-TUG score over time (IBM-FRS: Time X rho estimate = −3.32, $p = 0.034$; m-TUG: Time x rho estimate = 1.95, $p = 0.042$) (Fig. 8A). Of note, the model predicts that a shift in rho from the highest value observed in the study to the lowest from week 16 to week 32 would lead to an improved IBM-FRS and m-TUG (Fig. 8B). This suggests that the reduction in rho by pioglitazone observed in patients with mild disease may have improved outcomes in patients with high baseline rho values.

## Discussion

This clinical trial tested the hypothesis that IBM is associated with metabolic dysfunction and that pharmacologically targeting mitochondria with pioglitazone could improve muscle metabolic deficits in this population. We observed a trend of increased *PPARGC1A* gene expression after pioglitazone, although this difference did not reach significance ($p = 0.099$). Fewer patients than anticipated enrolled in the interventional portion of the trial in part due to the COVID-19 pandemic, and therefore the study may have been underpowered for this primary outcome. While changes in *PPARGC1A* transcript levels did not significantly differ, we did observe significant upregulation of its downstream activity, including activation of *PPARGC1A*-containing KEGG pathways such as AMPK signaling and mitochondrial-related pathways such as oxidative phosphorylation. A subset of patients showed metabolic changes during treatment, and in an exploratory analysis, these changes were associated with a slower decline in IBM-FRS and m-TUG scores. While the short follow-up period likely precluded detection of significant clinical differences, our findings

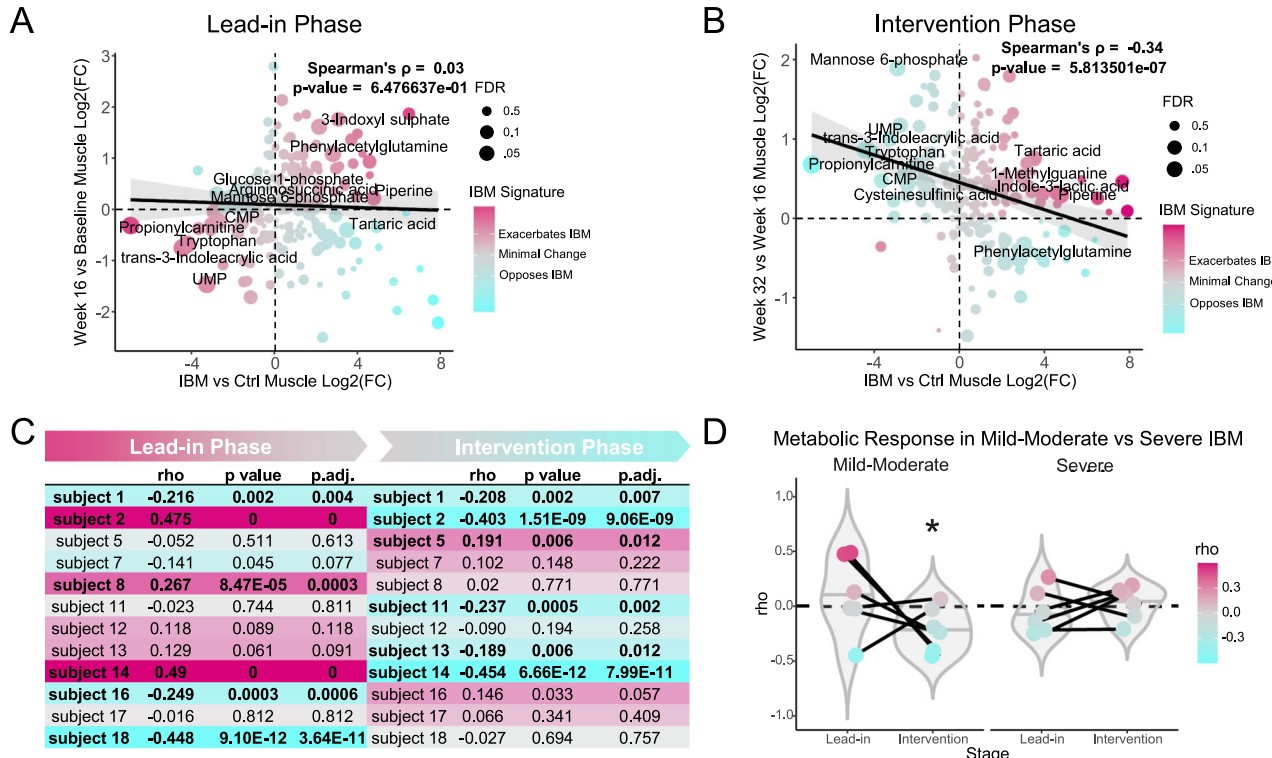

**Fig. 7 | Changes in the muscle metabolic signature in inclusion body myositis (IBM) during the lead-in and pioglitazone intervention periods.** Correlation analyses comparing the metabolic signature in muscle from 12 patients with inclusion body myositis (IBM) at **A** the end of the lead-in period (week 16) versus baseline, and **B** the end of the intervention period with pioglitazone (week 32) versus week 16. Metabolites in cyan indicate a shift away from the IBM metabolic signature, while metabolites in fuchsia indicate a shift in the same direction as the IBM signature. The significance of the correlation was determined using Spearman's correlation coefficient. Black line is a linear regression with 95% CI for visualization purposes. **C** Spearman's correlation coefficient for each patient during the lead-in period and the intervention period, with cyan indicating a shift away from the IBM signature. **D** Change in Spearman's correlation coefficient with pioglitazone, stratified by severe versus mild-moderate disease based on muscle ultrasound findings. Two-way ANOVA with Tukey HSD, *$p < 0.05$. UMP uridine monophosphate. CMP cytidine monophosphate. Rho Spearman's rho. Source data are provided as a Source data file.

provide a basis for larger studies to investigate the role of metabolic dysfunction in IBM.

The availability of baseline IBM specimens acquired in this trial enabled us to explore differences in metabolites in muscle and sera from patients with IBM and healthy controls. We observed significant differences across multiple metabolic pathways, underscoring the complex metabolic derangements in IBM, mirroring the findings of other metabolomics studies in IBM[26]. Many of the metabolic aberrations in IBM muscle tissue converge on the TCA cycle, which is central to producing NADH for the mitochondrial respiratory chain as well as providing metabolic precursors and intermediates for many other key metabolic processes of the cell[27]. Our data suggests that there is a distal impairment in the TCA cycle, which potentially arises from two mechanisms. One could be the diversion of glutamate to alpha-ketoglutarate and reductive (backwards) flow of the TCA cycle towards citrate, a mechanism known to occur under conditions of mitochondrial stress[28,29]. The second potential contributor to distal TCA cycle dysfunction is a decrease of propionyl-CoA, which feeds the TCA cycle at succinyl-CoA. The etiology of the reduced propionyl-CoA is not clear. Interestingly, pyrimidines can also contribute to replenishing intermediates of the TCA cycle during times of bioenergetic stress, and possibly this could account for the decreased pyrimidine content in IBM tissue[18]. Ultimately, isotopic flux analysis in patient-derived cells will help to clarify these metabolic dysregulations.

Treatment with pioglitazone was associated with modest, broad shifts in the muscle metabolome toward the profile of healthy controls and away from the IBM metabolic signature. This finding was not attributed to specific metabolites significantly altered by pioglitazone but rather broad metabolic changes across multiple pathways that shifted in a favorable direction. It is important to note that these analyses were correlative, and we do not know which, if any, of the altered metabolites are drivers of disease. Notably, these effects were confined to muscle, with no corresponding changes observed in serum, suggesting that pioglitazone's primary target in IBM is muscle tissue and its effects were not mediated by systemic alterations. The metabolic response to pioglitazone was somewhat heterogeneous, but the subset of patients with metabolic improvement had a slower decline in their IBM-FRS and m-TUG over time. These results provide preliminary evidence that pioglitazone may modulate metabolic pathways in IBM muscle, but larger controlled trials are needed to determine whether this translates into meaningful clinical benefit.

Pioglitazone is known to have diverse mechanisms of action. Our transcriptomic data indicate that pioglitazone upregulates PPAR and AMPK signaling, and that the effects in IBM muscle tissue are at least partially mediated through enhanced metabolic pathways and decreased inflammation. This is consistent with the known effects of pioglitazone in muscle from insulin-resistant individuals[13,30]. Pioglitazone is also known to improve skeletal muscle fatty acid metabolism and reduce intramyocellular lipid content[31], alongside its antioxidant properties and ability to stimulate autophagy, which may further contribute to its effects on the metabolome[32]. We observed increased gene expression in pathways related to amino acid metabolism and fatty acid degradation with pioglitazone, providing evidence that other mechanisms may contribute to its favorable metabolic effects in IBM. However, further research is needed to determine whether these

**Table 1 | Clinical outcomes at each of the four study visits**

| | Patients who completed all 4 study visits (n = 8) | | | | Patients who completed 3 study visits (n = 13) | | | | |
|---|---|---|---|---|---|---|---|---|---|
| | Baseline | Week 16 | Week 32 | Week 48 | Baseline | Week 16 | Week 32 | Estimate | p-value |
| Number of falls in prior 16 weeks | | 0.62 ± 0.77 | 0.69 ± 1.1 | 0.69 ± 0.95 | | | | 0.08 | 0.82 |
| Manual muscle testing knee extensor | 8.25 ± 2.12 | 7.88 ± 2.23 | 7.75 ± 3.11 | 7.62 ± 2.56 | 8.69 ± 1.75 | 8.53 ± 1.94 | 8.0 ± 2.8 | 0.19 | 0.76 |
| Knee extensor dynamometry (lbs) | 21.3 ± 13.9 | 20.9 ± 15.8 | 16.9 ± 13.7 | 20.4 ± 18.0 | 26.2 ± 14.5 | 22.6 ± 12.9 | 22.4 ± 16.7 | 5.6 | 0.10 |
| M-TUG (seconds) | 10.62 ± 3.27 | 10.49 ± 1.86 | 10.98 ± 2.38 | 11.71 ± 2.99 | 9.20 ± 3.26 | 9.20 ± 2.33 | 9.50 ± 2.94 | 0.19 | 0.72 |
| 6-min walk test (feet) | 1305.88 ± 237.40 | 1266.5 ± 135.01 | 1234.25 ± 172.17 | 1155.50 ± 162.08 | 1369.77 ± 269.13 | 1361.39 ± 301.62 | 1394.08 ± 362.48 | -46 | 0.45 |
| Hand-grip strength (lbs) | 30.31 ± 22.36 | 28.5 ± 21.24 | 27.25 ± 21.47 | 28.5 ± 23.41 | 29.88 ± 18.95 | 28.19 ± 17.99 | 27.27 ± 17.49 | 3.06 | 0.06 |
| IBM-FRS | 28.88 ± 3.91 | 28.88 ± 3.98 | 28.63 ± 3.58 | 28.0 ± 4.87 | 30.46 ± 4.22 | 30.46 ± 4.10 | 30.23 ± 4.13 | -1.08 | 0.21 |
| Creatine Kinase (U/L) | 497.63 ± 384 | 718.88 ± 584 | 539.86 ± 404 | 463 ± 264 | 482.77 ± 387 | 613 ± 520 | 506 ± 445 | -179 | 0.15 |
| Composite Fi2 | 5.7 ± 1.9% | 5.6 ± 2.2% | 5.2 ± 2.0% | 5.8 ± 2.2% | 6.4 ± 2.0% | 6.7 ± 2.3% | 6.1 ± 2.3% | 0.004 | 0.44 |

Outcome measures included knee extensor strength measured by manual muscle testing (Kendall Score) and muscle dynamometry, the Modified Timed Up and Go Test (m-TUG), 6-min walk test, hand-grip strength, Inclusion Body Myositis Functional Rating Scale (IBM-FRS), creatine kinase level (upper limit of normal 170 U/L), Composite Functional Index-2 (Fi-2), and number of self-reported falls during the preceding 16 weeks. Results are reported as the mean ± standard deviation. 5 patients did not come in-person for the last study visit due to the COVID-19 lock-down, so the results are reported separately for the cohort of patients who completed all 4 study visits (n = 8) and the cohort who completed the first 3 study visits (n = 13). A creatine kinase level and number of falls were able to be measured remotely for all participants at all 4 study visits. A linear mixed effects model was used to determine the significance of time x treatment [pio].

metabolic effects are specific to IBM or whether they occur in other conditions as well.

The effect of pioglitazone on the muscle metabolome tended to be more pronounced in patients with mild-moderate disease, suggesting that early disease may be more amenable to metabolic therapeutic intervention. Patients with a shift in their muscle metabolome away from the IBM metabolic signature had an improved disease trajectory in 2 out of the 7 outcome measures as compared to patients without such metabolic changes. Collectively, these observations provide preliminary data suggesting that pioglitazone may modulate metabolic pathways in early IBM, but whether this affects disease biology is unknown.

This clinical study had several limitations, including a small sample size, the absence of a placebo control arm, and a short intervention period with relatively insensitive clinical outcomes, limiting our ability to detect small differences in disease trajectory. The controls were not matched by sex and race, and we used archived open muscle biopsy specimens, which were obtained using different protocols compared to the IBM patients. Likely a consequence of the small sample size, models with adjustments for sex tended to display worse model fit metrics (Akaike and Bayesian information criterion), potentially indicating overfitting. Analyses were therefore performed without covariate adjustments beyond accounting for individual variability in repeated measures. The COVID-19 pandemic also hindered our ability to obtain complete clinical outcome measures at week 48 and led to higher patient discontinuation, which underpowered our study. Despite these challenges, our study design had notable strengths, such as implementing a lead-in period to serve as a comparator for each patient given the heterogeneity of this disease, utilizing muscle imaging to better characterize disease severity, and collecting paired blood and muscle samples at multiple longitudinal time points that allowed us to explore the metabolic dysfunction in IBM.

Our findings provide insights into the metabolic complexities underlying IBM and identified numerous dysregulated metabolic pathways in muscle. Although the primary endpoint was not met, pioglitazone treatment was associated with measurable transcriptomic and metabolic changes in IBM muscle that shifted its profile toward that of healthy controls. These exploratory findings support further investigation of muscle metabolism as a potential area of study in IBM.

## Methods

### Ethics statement

This single-arm trial was approved by the Johns Hopkins Institutional Review Board (IRB00130996) and was conducted in accordance with the principles of the Declaration of Helsinki. All study participants provided written informed consent and were provided modest financial compensation for their time and travel.

### Study population and design

The study was pre- registered on ClinicalTrials.gov (Identifier: NCT03440034) on February 21, 2018. The trial was conducted between June 2018 through June 2020 at the Johns Hopkins Myositis Center. The study protocol, along with a summary of changes to the protocol, can be accessed in the supplemental information file. Eligible patients included adults 50 years and older (to enrich for those with typical IBM features) with a diagnosis of clinico-pathologically defined or clinically defined IBM by ENMC criteria[33], were able to ambulate at least 20 feet with or without the use of an assistive device, and could rise from a standard chair (seat height 18 inches) on their own with the use of armrests if needed. Control muscle specimens were obtained from the Johns Hopkins Neuromuscular Biobank. These samples were collected from patients undergoing muscle biopsy for clinical evaluation. We selected available biopsies from older individuals whose pathology reports were interpreted as normal and who were ultimately

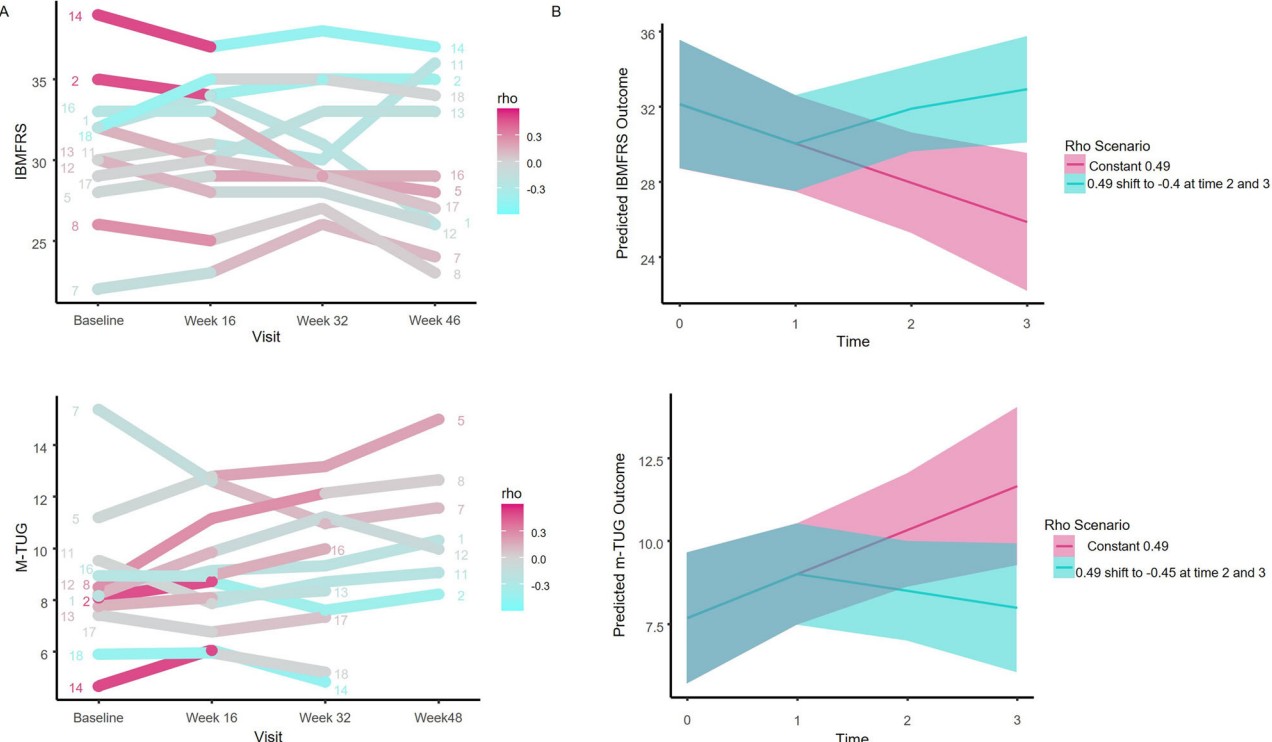

**Fig. 8 | Metabolic response in muscle and clinical outcome measures.**
**A** Individual metabolic correlation scores (rho) for each patient ($n$ = 12) during the lead-in period (baseline to week 16) and the intervention period (week 16 to week 32) in relation to the IBM-FRS and m-TUG. A mixed-effect model was used to determine the impact of metabolic response (rho) on each clinical outcome measure, revealing a significant rho x time effect for IBM-FRS and m-TUG. Fuchsia indicates a high rho reflecting increasing similarity to IBM metabolic signature, cyan reflects a low rho indicating reduced similarity to IBM metabolic signature.

**B** Model prediction of the effect of a shift in rho on IBM-FRS and m-TUG from the highest value observed in the study to the lowest value from week 16 to week 32. cyan reflects predicted function for subject with IBM like signature of rho = 0.49, which shifts away from IBM signature at time 1 to a rho of −0.45, while fuchsia reflects predicted values for an individual with a consistent IBM like signature of rho = 0.49. Bands reflect 95% confidence interval. IBM-FRS Inclusion Body Myositis Functional Rating Scale. M-TUG modified timed up and go. Source data are provided as a Source data file.

not diagnosed with, nor suspected to have, a neuromuscular disorder. Serum samples were obtained from healthy controls as part of a separate IRB-approved study for healthy donors and excluded anyone who was pregnant or had a history of cancer, autoimmune disease, tuberculosis, HIV, or hepatitis infection.

The study schema is depicted in Fig. 1 and more details are included in supplementary information. All participants who met the inclusion criteria completed a baseline visit and then were followed without any intervention until their second visit, 16 weeks later (lead-in phase) to serve as a self-comparator. Pioglitazone was started at week 16, and all patients self-administered 30 mg of pioglitazone by mouth daily for 2 weeks and then increased the dose to 45 mg daily. The study drug was continued for a total of 32 weeks (intervention phase), and patients were evaluated at week 32 and week 48. Patients brought in their empty pill bottles to each study visit and were questioned about adherence. At the baseline visit, week 16, and week 32, a needle muscle biopsy from the vastus lateralis was performed, functional measures were obtained, and serum was collected. At week 48, only serum and functional measures were obtained.

The primary, secondary, and exploratory outcomes were assessed at predefined timepoints. Primary and secondary transcriptomic and metabolic outcomes were evaluated by comparing the 16-week lead-in period (baseline to week 16) with the first 16 weeks of pioglitazone treatment (week 16 to week 32). Clinical secondary outcomes were assessed longitudinally at baseline, week 16, week 32, and week 48 when available. The trial was terminated earlier than planned due to the COVID-19 pandemic, which limited completion of later study visits and long-term outcome assessments.

## Safety monitoring
Safety labs were drawn at each visit, including a fasting blood glucose, creatinine, liver enzymes, and a pro-B-type natriuretic peptide (pro-BNP). Adverse events were monitored and reported in a standardized manner using the Common Terminology Criteria of the National Cancer Institute V4.03, with the lead investigators (BA, JA) determining their relatedness to the drug.

## Outcome measures
**Primary outcome.** The primary outcome was the change in expression of *PPARGC1A* and other oxidative phosphorylation and mitochondrial pathway genes, comparing the 16 week pioglitazone treatment phase to the 16 week lead-in period.

## Secondary outcomes
Secondary outcomes included the change during the treatment phase compared to the lead-in period for metabolites in key energy pathways and for several clinical outcome measures described below. All clinical measures were assessed at each study visit using standardized procedures. licensed physical therapists (AM, RP) and occupational therapists (MM) underwent training prior to trial initiation to standardize assessments.

i.   *Manual muscle testing:* the Kendall 0–10 scale was used to assess strength in the knee extensors[34]. Scores were determined by the examiner's assessment of resistance against gravity and manual pressure, with higher scores indicating greater muscle strength. Strength was measured in the bilateral knee extensors 3 times and then averaged.

ii.  *Quantitative muscle dynamometry:* A handheld dynamometer (Hoggan Scientific MicroFET®2) was used to assess quantitative muscle strength in the knee extensors. Participants were seated with their knees at 90° of flexion. The dynamometer was placed on the distal part of the lower leg, just above the ankle, and manual resistance was applied gradually until the dynamometer recorded a measurement in pounds. Strength was measured in both lower extremities and then averaged.

iii.  *Hand-grip strength:* Grip strength was measured using a Jamar hydraulic hand dynamometer following standardized procedures[35]. Participants were seated with the elbow flexed at 90° and the forearm in a neutral position. The test was performed on the standard (second) handle setting, and the best of three trials for each hand was recorded. Grip strength for each side was then averaged.

iv.  *Modified Timed Up and Go Test (m-TUG):* Participants were instructed to rise from a standard-height chair (seat height 18 inches), walk a distance of 3 meters at a comfortable pace, turn around, return to the chair, and sit down[36]. The time required to complete the task was measured. Participants were permitted to use armrests to assist with standing if needed. A single trial was performed.

v.  *Six Minute Walk Test:* Participants were instructed to walk as far as possible for 6 min along a marked, flat indoor hallway. Standardized verbal encouragement was provided. Participants were allowed to slow down, stop or rest as needed but were instructed to resume walking as soon as they were able. Total distance walked was recorded in feet. Participants were allowed to use assistive devices such as canes or walkers, and the use of these devices was documented[37].

vi.  *IBM-Functional Rating Scale (IBM-FRS):* This questionnaire was administered by the physical therapist and assessed functional status across 10 domains relevant to daily living in IBM. Each item is scored on a 0–4 scale, with high scores indicating better function. The maximum total score is 40[38].

vii.  *Composite Functional Index-2:* The FI-2 measures the number of repetitions that can be performed over a 60-s period for each of seven muscle groups: shoulder flexion, shoulder abduction, neck flexion, hip flexion, knee extension, ankle dorsiflexion, and heel lifts. Participants were instructed to perform each movement as many times as possible within the time limit[39]. The percent of completed repetitions was recorded for each muscle group and averaged to derive a composite Fi-2.

viii.  *Creatine kinase (CK) level:* this was measured in a commercial laboratory.

ix.  *Number of Falls:* the number of self-reported falls during the preceding 16 weeks was recorded.

## Muscle ultrasound

A GE Logiqe or S8 machine with a linear probe was utilized with standardized settings as previously described[22]. All patients had an ultrasound (US) of the vastus lateralis at the screening visit to ensure that there was viable muscle for biopsy, which was an inclusion criteria for this trial. Muscle US images of the flexor digitorum profundus (FDP), vastus lateralis (VL), and rectus femoris (RF) muscles were acquired at entry into the study to assess muscle quality and stratify severity based on prior studies showing them as discriminating muscles in IBM[21,22]. Each muscle was assessed for atrophy and echointensity using the Heckmatt scale (1–4)[40]. The highest echointensity score for each muscle on either side was taken for each patient and added to create a sum score. To have a clear distinction of a more advanced phenotype, we defined severe muscle disease as a sum score >10 for echointensity with atrophy.

## Muscle biopsies and biospecimens

Needle muscle biopsies of the vastus lateralis were performed at the baseline visit, at week 16, and week 32, using the same side for each patient. Muscle biopsies were performed as previously described using a Pro-Mag Ultra Automatic Biopsy instrument (Argon Medical Devices) with site for biopsy located by ultrasound[41]. Specimens were immediately frozen in liquid nitrogen or stored in RNA later (Thermo Fisher). Control muscle specimens obtained from the Johns Hopkins Neuromuscular Biobank were obtained by open surgical biopsy of the rectus femoris, biceps, or deltoid for clinical indications and were immediately flash-frozen. Sample processing and metabolomic analysis of control and IBM specimens occurred in parallel.

## Metabolomics extraction and analysis

Untargeted metabolomics was performed on all serum and muscle samples at Gigantest (Baltimore, MD). 20 muscle samples and 25 serum samples were analyzed. Controls consisted of pooled samples and were dispersed throughout the run between every 10 samples (total of 5 controls). Number of technical and/or biological replicates was 1. The metabolite extraction of serum and muscle samples was done by adding HPLC-grade methanol to achieve a 80% (vol/vol) final methanol concentration. Muscle samples were homogenized to ensure metabolite extraction. Both types of samples were then subjected to speed vacuum and lyophilization to ensure removal of the methanol-water mixture. A 50% (vol/vol) acetonitrile solution was then used to resuspend the lyophilized metabolite samples for LC-MS data acquisition. The LC-MS system consisted of a Thermo Scientific Q Exactive Plus Orbitrap Mass Spectrometer with a Vanquish UPLC. Metabolomics data acquisition of the metabolite-extracted samples was performed as previously described[42]. Briefly, reverse phase chromatography was performed using a Sigma HSF5 column (15 cm × 2.1 mm, 3 μm) with a 15-min total run time. Mobile phase is 0.1% formic acid in water, and mobile phase B is 0.1% formic acid in acetonitrile. Full MS and Full MS/ddMS2 were acquired. The column was kept at 35 °C while the autosampler was kept at 4 °C. Data annotation was performed using Thermo Scientific Compound Discoverer®, XCalibur®, TraceFinder® software, and was achieved through fragmentation pattern matching with the mzCloud database and/or accurate m/z match within 5 ppm. Chromatography peak integration was reviewed, and peaks were manually integrated if required. Final peak intensities were normalized to the sample protein concentration measured with a BCA-based assay. The raw intensities of discovered metabolites were normalized according to the protein concentration of the samples.

## RNA library preparation and sequencing

RNA was isolated from muscle tissue using the RNeasy Plus Kit (Qiagen). Samples were checked for quality/quantity on the Fragment Analyzer and Qubit prior to library prep. NEBNext Poly(A) Magnetic Isolation Module (NEB #E7490) and NEBNext Ultra II RNA Library Prep Kit for Illumina (Cat# E7775) were used to generate libraries per manufacture instructions. The Fragment Analyzer was used for quality control to ensure adequate concentration and appropriate fragment size. The resulting library insert size was 200–500 bp with a median size around 300 bp. Libraries were barcoded using unique dual indexing (E6440S) and pooled for sequencing. Pooled libraries were sequenced on an Illumina NovaSeq6000 instrument using standard protocols for SP 2 × 100 bp paired-end sequencing. 13 samples were multiplexed for an estimated 558109295 reads per sample. Demultiplexed FASTQ files were generated using bcl-convert v3.7.5.

## Statistical analysis

Baseline demographic, clinical, and laboratory characteristics were compared between the IBM and control groups using descriptive statistics. All statistical tests were 2-sided when applicable, and a $p$-value $< 0.05$ was considered significant. Differences in continuous variables were assessed with the Wilcoxon rank-sum test, and categorical variables were compared using Fischer's exact test. RNA

sequencing data were analyzed with DESeq2 to determine the genes present in each patient, their expression levels, and the differences between expression levels in the muscle pre- and post-treatment. Gene set enrichment analysis using KEGG pathways was performed with the clusterprofiler package (Version 4.6.2). Pathway network analysis was performed with the aPEAR package (Version 1.0.0), with minimum cluster size set to 2, and cluster method "hier". Cluster labels were determined using the default "pagerank" setting to find the most connected node, with limited adjustments for improved consistency between analyses. Ingenuity pathway analysis was performed using the log2 fold change values corresponding to lead-in and treatment-phase to perform "upstream regulator analysis" of *PPARGC1A* target genes in each phase. Change in *PPARGC1A* normalized counts was compared between the lead-in (baseline to week 16) vs the treatment (week 16 to 32) phase with a paired t-test using the "t.test" function.

Metabolomics data were analyzed using MetaboAnalyst and R. Epinephrine was detected in muscle samples due to use with anesthetic for biopsy and was therefore removed from analysis due to the non-physiological nature of its abundance. Data were log transformed, and samples normalized to the median. PCA plots were generated using the prcomp function and plotted with the autoplot function of the ggfortify package (Version 0.4.17). Differentially abundant metabolites were determined using lmFit and eBayes function (Version 3.54.2) and plotted with ggplot2 (Version 3.5.1). Pathway analysis was done using MetaboAnalyst and the KEGG pathways, with a custom background library based on all detected compounds in the experiment.

To test the effect of changes in the IBM metabolic signature (evaluated as rho), Spearman's rho was calculated for each patient using their log2FC for each metabolite from baseline to week 16 and from week 16–32 compared to the log2FC of all IBM patients compared to controls. Each patient's rho for baseline to week 16 was used for those two time points, while their rho for week 16–32 was taken as representative for week 32 and 48. A linear mixed effects model, "outcome -1+Treatment*Time + (1+Time|ID)", was used to calculate the effect of rho on outcomes. To define metabolic "responder" status, we compared each participant's rho during the lead-in and treatment phases using jackknife estimates of the confidence intervals, implemented via the SpearmanCI package. Participants were classified as responders only if there was no overlap in the confidence intervals between the two phases, indicating a significant within-patient change. The subgroup assessment of metabolic response by muscle US severity was performed using a two-way mixed ANOVA with the "aov" function in R. A "leave-one-out" sensitivity analysis was then performed, repeating the ANOVA with each individual left out of one analysis to identify possible influential patients.

Integrated transcriptomic-metabolomic pathway analysis was performed using the "multiGSEA" package with KEGG pathways.

To evaluate the effect of pioglitazone treatment on outcomes, data were analyzed using a linear mixed effects model with the *lmer* function in the lme4 package (Version 1.1.35.1) in R (Version 4.2.0). Model equations were "outcome -1+Treatment*Time + (1+Time|ID)" for all outcomes besides hand grip and knee dynamometry, which had the random slope for time removed to improve model fit.

### Reporting summary
Further information on research design is available in the Nature Portfolio Reporting Summary linked to this article.

## Data availability
De-identified participant-level clinical, metabolomic, and transcriptomic data are available via the Vivli Center for Global Clinical Research for non-commercial research (https://doi.org/10.25934/PR00011654). Due to privacy laws, access requires proposal review and a Data Use Agreement through Vivli (https://vivli.org), typically completed within 4–8 weeks. Source data for figures are provided in the Supplementary Information/Source Data file. Source data are provided with this paper.

## Code availability
The code supporting the findings of this study are available in the Vivli repository (https://doi.org/10.25934/PR00011654).

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

## Acknowledgements

Library prep and Illumina sequencing were conducted at the Genetic Resources Core Facility, Johns Hopkins Institute of Genetic Medicine, Baltimore, MD. This work was funded by the Ira T. Discovery Fund (JA), Peter and Carmen Lucia Buck Foundation Myositis Discovery Fund (BA), the Peter Frampton Myositis Research Fund (LCS), the Jerome L. Greene Foundation (JA, ED, CM, BA), and the National Institutes of Health (NIH) Grants 1S10OD025226-01 (A.L), T32AR048522 (BA and MRB), 2T32AG058527-06A1 (MRB), K23AR075898 (CM), K23AR073927 (JJP), K08AR077732 (ET), and K23AI180356 (BA).

## Author contributions

Design of research study- B.A., J.A., A.L., and L.C.S. Conducting experiments- B.A., J.A., A.L., C.Z., and P.K. Data acquisition—B.A., J.A., L.C.S., J.P., C.M., E.T., T.C., E.D., T.L., A.L., C.Z., P.K., C.S., R.P., M.M., and A.M. Data analysis- B.A., J.A., M.B., C.Z., A.L., and H.V. Manuscript preparation and writing—B.A., M.B., J.A., L.C.S., J.P., C.M., E.T., T.C., E.D., T.L., A.L., C.Z., P.K., C.S., R.P., M.M., A.M., and H.V.

## Competing interests

E.D. is currently an employee of AstraZeneca. A.L. is the founder and CEO of Gigantest. The remaining authors declare no conflicts of interest.
