## [Transparent Peer Review file · Nature Communications]

Reversing Muscle Metabolic Abnormalities in Inclusion Body Myositis with Pioglitazone: A Single Arm Clinical Trial

Corresponding Author: Dr Brittany Adler

Version 0:

Reviewer comments:

Reviewer #1

(Remarks to the Author)

The authors performed a small open-label study of pioglitazone in IBM. I have the following concerns:

1. The authors mention that "Eligible subjects included adults 50 years and older with a diagnosis of clinico-pathologically defined or clinically defined IBM by ENMC criteria (Rose 2013) or data-driven criteria (DDC) (Lloyd 2014)."
1a) Why was the age cut-off set at 50 years when the threshold for the 2013 ENMC criteria is 45 years, and this was further reduced to 40 years in the 2024 ENMC criteria?
1b) The category of clinically defined IBM (mentioned by the authors in eligibility criteria) does not exist in either Rose 2013 or Lloyd 2014. There is inconsistency in the definition of the recruited population.
2. I could not find a ClinicalTrials.gov registration number. Was the study not registered?
3. The study protocol and statistical analysis plan (SAP) were not provided. Could these be made available if they exist? Were all the analyses presented pre-specified in the protocol and SAP? What are the versions and dates of the protocol and SAP, and when did the database lock occur?
4. When was the first patient's first visit, and when was the last patient's last visit?
5. The eligibility criterion states that participants "...could rise from a chair on their own." Were the dimensions of the chair standardized? Could the chair have armrests?
6. The inclusion criterion "Must have viable quadriceps muscle suitable for biopsy as seen on ultrasound and interpreted by the investigator" is repeated in the Supplemental Methods (inclusion criteria 5 and 8). One item can be removed.
7. What was the upper limit of normal for CK?
8. How was "Malnutrition or malabsorptive syndrome defined as less than 0.8 g protein/kg/day" evaluated or defined?
9. Why is "bladder cancer" specifically listed as an exclusion criterion when "History of cancer within the last five years, other than local basal or squamous cell carcinoma," is already included?
10. There are significant gender and racial differences between IBM muscle/sera and control muscle/sera. Could these differences have influenced the results? Why was matching not performed?
11. Was any method of drug accountability used? This is not mentioned.
12. The manuscript states, "At the baseline visit, week 16, and week 32, a needle muscle biopsy from the vastus lateralis was performed." Were the biopsies taken from the same side and area of the vastus lateralis? Given that IBM is a patchy and asymmetrical disease, repeated biopsies may be problematic. If subsequent biopsies were taken from the same area, results could be affected by scar tissue. If taken from different locations, results could be influenced by varying levels of inflammation, degeneration, fat deposition, and mitochondrial changes, as these features are heterogeneously distributed in

IBM muscle.

13. Which methods and devices were used for assessing “knee extensor strength by quantitative dynamometry” and “hand-grip strength using handheld dynamometry”? Which side of the body was assessed? If both sides were assessed, was the average value used? This is not explained in the manuscript

14. Regarding the “modified timed up and go” (m-TUG) test, how did the authors handle patients who were unable to perform the test, mathematically/in the change analysis?

15. The manuscript states, “Certified physical (AM, RP) and occupational therapists (MM)...” What does “certified” mean in the context of this trial? Which entity provided the certification? Did this include specific training on trial outcome assessments?

16. The primary outcome is defined as “the change in PPARGC1A gene transcription measured by RNA sequencing after pioglitazone (week 32) compared to before treatment (week 16),” which corresponds to the intervention phase. However, the secondary outcome measures are described as changes “from baseline,” with baseline defined as week 0 (16 weeks before pioglitazone initiation, as per Figure 1). Why was a different time frame used for the secondary outcomes (week 0 to week 32) than for the primary outcome (week 16 to week 32)? This inconsistency is problematic.

17. The manuscript states, “Baseline demographic, clinical, and laboratory variables were evaluated with descriptive statistics, and differences were evaluated using the Wilcoxon rank sum test for continuous variables and Fisher’s exact test for categorical variables.” This sentence is unclear. What differences were being compared? It reads as though baseline values are being compared (e.g., between IBM and healthy subjects), in which case a paired test would not be appropriate.

18. The manuscript states, “Genes with a ≥ 1.5 log₂ fold change in expression were considered to be significantly differentially expressed (at $P < 0.05$).” Given the large number of genes assessed, why was a more stringent p-value cut-off not applied? Why was a correction method such as Bonferroni not used?

19. The manuscript states, “Epinephrine was detected in muscle samples.” Could this have influenced the expression of other metabolites?

20. The analysis of changes in the IBM metabolic signature is difficult to interpret. How can a correlation between two pre-treatment values (baseline and week 16) be compared to a correlation between week 16 and week 32? How does this inform the treatment effect? This approach is unconventional. Why were week 16 and week 32 results not compared directly, as was done for the primary endpoint?

21. Does “average duration of IBM” refer to the time since symptom onset or since diagnosis?

22. How was control muscle obtained? Was it taken from the vastus lateralis using the same technique as in IBM patients?

23. The manuscript states, “Untargeted metabolomics was performed on the 15 remaining samples and compared to five healthy control muscle specimens.” These sample sizes are extremely low for metabolomics studies, raising concerns about the validity of the results and the potential influence of outliers. The same concern applies to the serum sample analyses.

24. The manuscript states, “Patients with a US sum score ≥ 10 were characterized as having severe disease (n=7), and patients with a US sum score < 10 were characterized as having mild-moderate disease (n=6).” Has this cut-off ever been validated? What was the rationale for choosing it?

25. The manuscript states, “Using a linear mixed-effects model, we observed increased PPARGC1A gene expression after pioglitazone, but this did not reach statistical significance ($p=0.07$).” Please clearly state that there was no treatment effect of pioglitazone on the primary endpoint rather than focusing on “trends.” The same comment applies to the discussion section.

26. The lack of a placebo group is a major limitation of this study. This should be further emphasised. Why was a placebo group not evaluated?

27. Are the observed metabolomic changes specific to IBM? Would similar changes be expected in other conditions or even in healthy individuals receiving pioglitazone?

28. The statistical methods section is unclear. Some statistical tests (e.g., Two-way ANOVA with Tukey HSD, mentioned in Figure 7) are referenced in figures but not described in the methods section.

29. The manuscript states, “These metabolic responders tended to have earlier and less severe disease as stratified by US (Figure 4A, 4B, 7D).” However, these data are not convincing. Figure 7D shows only five patients in the mild-moderate group; if one “red dot” patient is removed, the effect would disappear, just as it is absent in the advanced group.

30. This is a negative study in IBM patients. Highlighting a single patient with subjective “clinical improvement” is inappropriate. For some outcome measures, the reported variation for this patient falls within the expected measurement error. Furthermore, given that this patient was obese and had metabolic syndrome, any observed “improvement” may simply reflect better cardiovascular and metabolic health rather than an IBM-specific effect.

31. The manuscript states, "In a subset of patients, mostly with mild-moderate disease, a demonstrable metabolic response was seen in IBM muscle, which was associated with a slower decline in IBM-FRS and m-TUG scores." Please tone down this statement, as the data are not convincing (see previous comments).

32. The statement, "These results provide initial evidence that pioglitazone can partially reverse the metabolic abnormalities of IBM muscle and that this reversal can translate into improved clinical outcomes for a subset of patients," should be toned down, as the data do not convincingly support this conclusion.

33. Table S2 states that comparisons between baseline values of IBM patients and controls were made using the paired Wilcoxon test. This is inappropriate because these are independent samples.

34. Table S7 should be included in the main manuscript, as it clearly shows that the study was negative for all assessed physical function, strength outcome measures, and CK. For each outcome measure, please present their units, if applicable.

Reviewer #2

(Remarks to the Author)

This study is an exploratory, single-arm clinical trial investigating the effects of pioglitazone in patients with inclusion body myositis (IBM), with metabolomic analysis of biopsied muscle as the primary endpoint. Additionally, the study compares the metabolomic and transcriptomic profiles of patients at baseline with those of healthy controls from a biobank, allowing for a clearer interpretation of treatment-induced changes in the context of disease-specific alterations.

Despite the small sample size, the study is well-designed, incorporating a lead-in period during which needle muscle biopsies were performed. This design adds significant value beyond a simple pre- and post-intervention metabolomic comparison. The manuscript is well-written and provides valuable insights. Although the effects of pioglitazone were not clearly detected—partly due to the exploratory nature of the trial—the findings contribute meaningfully to the design of future clinical trials for this debilitating disease. My specific comments are as follows:

Major Points

1. Figure 4 effectively illustrates the relationship between ultrasound (US) severity and muscle metabolome. The authors may also wish to examine the associations between US findings and IBM-FRS, as well as between metabolome profiles and IBM-FRS.
2. The study identifies a subset of responders based on muscle metabolic changes following pioglitazone treatment (Figure 7). To further elucidate the clinical significance of these metabolic alterations, the authors should compare motor performance and biomarker changes between responders and non-responders.
3. To strengthen the multi-omics approach, the authors should directly compare treatment-induced transcriptomic changes (Figure 6) with metabolomic alterations (Figure 7).
4. Similarly, a comparison between disease-related transcriptomic changes and treatment-induced transcriptomic alterations would provide additional valuable insights.

Minor Points

1. Was the muscle biopsy site in the control group the same as in the patient group (i.e., vastus lateralis)?
2. A more detailed discussion is needed regarding the possible causes of heart failure exacerbation in one patient.
3. There is an ethical concern regarding the re-initiation of pioglitazone in one patient. How did the ethical review committee justify this treatment outside of the study protocol?

Reviewer #3

(Remarks to the Author)

Reviewer #5

(Remarks to the Author)

General Comments

This was a small study which aimed to evaluate the impacts of pioglitazone on various outcomes and assessed the differences between healthy muscle specimens and serum samples with those from IBM patients. PPARGC1A expression increased at week 32 compared to week 16 (before treatment) $P=0.07$, indicating that pharmacologically targeting mitochondria with pioglitazone may improve metabolic deficits in this population.

For a single arm study, the "lead-in" phase was a helpful addition to assess the stability of the clinical outcomes assessed. The PPARGC1A gene expression (primary outcome) increased compared to pre-treatment timepoints however it is difficult to assess the extent of this since little information is reported about this outcome's results.

Small sample size and no controls does limit the study. More patients would have given the authors a better understanding of the effect pioglitazone. The lack of controls does mean that it is hard to assess a potential Hawthorne effect, however due to primarily biological outcomes (which can still be affected by being in a trial), and nature of the small/exploratory study

design I don't think this is a major flaw with the trial.

It is unclear if the statistical analyses were pre-specified prior to screening.

I agree with the authors' conclusions.

Main Comments

"Control muscle specimens were obtained from the Johns Hopkins Neuromuscular Biobank and were collected from patients with normal muscle tissue and no diagnosed neuromuscular disorders." and "Serum samples were obtained from healthy controls as part of a separate IRB-approved study for healthy donors." It is not clear what defines "normal muscle tissue or a healthy control?" Were these just samples from individuals without Inclusion body myositis or without a set of conditions? Were they randomly selected from the biobank/ healthy donors in the IRB-approved study or were they all samples the bank/study had at the time? Without this detail, it would be hard to reproduce the work.

"Using a linear mixed effects model, we observed increased PPARGC1A gene expression after pioglitazone, but this did not reach statistical significance ($p=0.07$)."

This is missing an effect size estimate and confidence interval. Since this is the primary outcome, I would have expected this to be clearly written. I think the interpretation of the P-value could be potentially less conservative with their (particularly considering the small sample size). They did observe that there was an increased PPARGC1A gene expression after pioglitazone. The plot for this (Figure 6A) also hides the confidence interval for the PPARGC1A Normalized Averages of Counts due to the black colour for the bars and the error bars.

Minor comments

4. "Model equations were "outcome $\sim 1+\text{Treatment}*\text{Time} + (1+\text{Time}|\text{ID})$ "

Did the authors consider adjusting for the baseline demographics they collected (age, sex, ethnicity, disease duration, biomarkers etc.)?

5. "The mean age of control muscle specimens was 60.6 ± 15.4 years and 4/5 (80%) were female ($n=5$). The control sera was slightly younger than the IBM sera (56.3 ± 5.7 years, $p=0.001$) and 7/10 (70%) were female ($n=10$)."

The "($n=5$)" and "($n=10$)" at the end of these sentences is slightly confusing as it is not clearly referring to something. I presume this is simply referring to the number of control sera and muscle specimens but this could be clearer.

6. "Using principal component analysis, we found that samples from healthy control and IBM muscle showed distinct clustering patterns, with a more variable clustering of IBM samples than healthy controls (Figure 2A)."

Though this is only 5 healthy controls so could be due to randomness/only having 5 controls.

7. "This stratification correlated with function and those with severe disease by US had lower IBM-FRS scores (28.57 ± 3.73 vs 33.33 ± 3.72)."

Are these recorded anywhere in a table/figure? I can't see them reported elsewhere.

8. "We did not observe any difference in the metabolic signature of severe versus mild-moderate IBM in sera (Spearman's $p=-0.13$, $p=0.06$) (Figure 4D)."

It is stated that the investigators did not observe "any" difference however there was a small difference observed. Whether this was due to chance or if there is a true difference is a slightly different matter.

9. "Despite notable agreement for a few key metabolites, such as propionylcarnitine and tryptophan which were both decreased in serum and muscle, the overall correlation between the metabolic profiles in muscle and serum was not significant (Figure 5E, Spearman's $p=0.06$)."

As with the statement regarding correlation between fold change between IBM vs. controls and severe vs mild IBM (and various other statements), it would be good for the P-value to be stated here also ($P=0.501$)

10. "After completing the clinical trial, the patient discontinued pioglitazone and his CK increased back to his pre-treatment baseline of 2055 units/L,"

CK units have been referred to as U/L throughout apart from this sentence.

11. "Although there was no statistically significant change in disease trajectory for the study population as a whole, one patient (subject 11) demonstrated a clinical improvement with pioglitazone. "

Some context around why this participant was followed up so closely after the end of the trial might be helpful since I can't see this mentioned in the methods (was this planned or just because of their promising CK levels during the trial). Were any other patients followed-up post 48 weeks? By chance you would expect someone to have a strong reduction in levels of one of these 7 outcome measures so inference from this patient's data should be limited (as I think it fairly is).

12. "despite missing the primary endpoint of increasing PPAR γ gene expression"

The choice of words here is slightly misleading/confusing – the primary endpoint was not missed. There was not as strong evidence as was hoped for but there was still some evidence.

Version 1:

Reviewer comments:

Reviewer #1

(Remarks to the Author)

I thank the reviewers for their responses, particularly their efforts to moderate some of their initial statements and conclusions, and to expand on the study's limitations. These revisions contribute to a more balanced and nuanced

perspective of the work, which also now seems aligned with the protocol.

One minor comment: The manuscript now states that "Eligible subjects included adults 50 years and older (to enrich for those with typical IBM features) with a diagnosis of clinico-pathologically defined or clinically defined IBM by ENMC criteria (18) or data-driven criteria (DDC) (19)". However, in the protocol only ENMC criteria are mentioned. This discrepancy still requires clarification.

Reviewer #2

(Remarks to the Author)

The authors addressed all the points I raised. I have no further concern.

Reviewer #3

(Remarks to the Author)

Reviewer #5

(Remarks to the Author)

Thank you for your active responses to these comments.

Point 8 was:

8. "We did not observe any difference in the metabolic signature of severe versus mild-moderate IBM in sera (Spearman's $p = -0.13$, $p = 0.06$) (Figure 4D)."

It is stated that the investigators did not observe "any" difference however there was a small difference observed. Whether this was due to chance or if there is a true difference is a slightly different matter.

Author response:

We have updated the language in the manuscript to be more precise, it now reads "...a significant difference...".

I appreciate the efforts to correct this however, it would be inappropriate to say a significant difference. Instead a better interpretation is "some difference..."

Open Access This Peer Review File is licensed under a Creative Commons Attribution 4.0 International License, which permits use, sharing, adaptation, distribution and reproduction in any medium or format, as long as you give appropriate credit to the original author(s) and the source, provide a link to the Creative Commons license, and indicate if changes were

made.

Thank you to the Reviewers for their valuable and insightful feedback on our manuscript. We have carefully considered each of the comments and have provided detailed, point-by-point responses. All changes made to the manuscript are clearly tracked. We believe these revisions have strengthened the manuscript, and we hope that the revised version meets the expectations of the Editor and Reviewers.

Reviewer #1 (Remarks to the Author):

The authors performed a small open-label study of pioglitazone in IBM. I have the following concerns:

1. The authors mention that “Eligible subjects included adults 50 years and older with a diagnosis of clinico-pathologically defined or clinically defined IBM by ENMC criteria (Rose 2013) or data-driven criteria (DDC) (Lloyd 2014).”

1a) Why was the age cut-off set at 50 years when the threshold for the 2013 ENMC criteria is 45 years, and this was further reduced to 40 years in the 2024 ENMC criteria?

We appreciate the Reviewer’s observation regarding the age cut-off. While the 2013 ENMC criteria set a threshold of >45 years, and the 2024 revision reduced this to >40 years, we selected an age cut-off of >50 years. At the time we designed the study, our intention for selecting this cutoff was to enrich the study cohort with individuals more likely to represent the classic clinical and pathological features of IBM. Patients under 50 may have more heterogeneous or atypical presentations, which could introduce diagnostic uncertainty or potentially confound interpretation of treatment response. To maintain a more homogeneous cohort for the purposes of this study, we opted for a more conservative age threshold. We have added a note in the manuscript to clarify this point.

1b) The category of clinically defined IBM (mentioned by the authors in eligibility criteria) does not exist in either Rose 2013 or Lloyd 2014. There is inconsistency in the definition of the recruited population.

Rose 2013 outlines both “clinico-pathologically defined IBM” and “clinically defined IBM” as part of the ENMC diagnostic criteria. For clarity, we have included Table 1 from Rose et al., 2013 below, which delineates these categories. In our opinion, the use of this terminology and classification in our eligibility criteria is consistent with this published framework.

Table 1
The ENMC IBM research diagnostic criteria 2011.

Clinical and laboratory features	Classification	Pathological features
Duration >12 months Age at onset >45 years Knee extension weakness \geq hip flexion weakness and/or Finger flexion weakness > shoulder abduction weakness sCK no greater than 15 \times ULN	Clinico-pathologically defined IBM	All of the following: Endomysial inflammatory infiltrate Rimmed vacuoles Protein accumulation [*] or 15–18 nm filaments
Duration >12 months Age at onset >45 years Knee extension weakness \geq hip flexion weakness and Finger flexion weakness > shoulder abduction weakness sCK no greater than 15 \times ULN	Clinically defined IBM	One or more, but not all, of: Endomysial inflammatory infiltrate Up-regulation of MHC class I Rimmed vacuoles Protein accumulation [*] or 15–18 nm filaments
Duration >12 months Age at onset >45 years Knee extension weakness \geq hip flexion weakness or Finger flexion weakness > shoulder abduction weakness sCK no greater than 15 \times ULN	Probable IBM	One or more, but not all, of: Endomysial inflammatory infiltrate Up-regulation of MHC class I Rimmed vacuoles Protein accumulation [*] or 15–18 nm filaments

2. I could not find a ClinicalTrials.gov registration number. Was the study not registered?

This study is registered in ClinicalTrials.gov. The title is “Study of Pioglitazone in Sporadic Inclusion Body Myositis” and the registration number is NCT03440034. This registration number is on the title page and we also added it to the Methods section.

3. The study protocol and statistical analysis plan (SAP) were not provided. Could these be made available if they exist? Were all the analyses presented pre-specified in the protocol and SAP? What are the versions and dates of the protocol and SAP, and when did the database lock occur?

We appreciate the reviewer’s request for additional detail regarding the protocol and analysis plan. We have provided the study protocol which contains the statistical analysis plan as supplementary materials.

In the revised manuscript, we now more clearly and explicitly report the pre-defined primary, secondary, and exploratory outcomes sequentially, as specified in the study protocol. We also updated the primary outcome description to match the protocol-approved definition: “Change in expression of PGC-1 α and other genes in the oxidative phosphorylation and mitochondrial function pathways.” Statistical analyses have been aligned with the pre-specified plan, using paired t-tests comparing changes in the lead-in and treatment periods. As specified in the protocol, pathway analysis was originally planned using Ingenuity Pathway Analysis (IPA). In the initial submission, we used clusterprofiler with KEGG pathways due to temporary loss of access to IPA. For the current revision, we obtained a new IPA license and reanalyzed the data, confirming consistent results across both platforms, with IPA identifying significant activation of PPARGC1A-regulated pathways during treatment. We believe this strengthens the evidence supporting target engagement and alignment with the pre-defined primary endpoint.

As this was a Phase I exploratory trial, no formal database lock was performed; however, the dataset used for analysis was cleaned and finalized on December 6, 2022.

4. When was the patient's first visit, and when was the last patient's last visit?

This trial was conducted from June 2018 through June 2020. These dates were added to the manuscript in the Methods section.

5. The eligibility criterion states that participants "...could rise from a chair on their own." Were the dimensions of the chair standardized? Could the chair have armrests?

We used the same chair for all participants throughout the trial. The chair was a standard chair with a seat height of 18 inches and armrests, in case the participant needed to use them. These details have been added to the manuscript.

6. The inclusion criterion "Must have viable quadriceps muscle suitable for biopsy as seen on ultrasound and interpreted by the investigator" is repeated in the Supplemental Methods (inclusion criteria 5 and 8). One item can be removed.

Thank you for catching this error. Inclusion criteria 8 has been deleted.

7. What was the upper limit of normal for CK? The upper limit of normal for CK is 170 units/L. This has been added to the legend for Table 1.

8. How was "Malnutrition or malabsorptive syndrome defined as less than 0.8 g protein/kg/day" evaluated or defined?

Thank you for the opportunity to clarify this criterion. The exclusion criterion which specified "malnutrition or malabsorptive syndrome defined as less than 0.8 g protein/kg/day," was adopted from other neuromuscular trials, although in practice this was difficult to assess and we relied on BMI and serum albumin levels as a surrogate (BMI < 18.5 kg/m² or serum albumin < 3.5 g/dL were considered to be indicative of malnutrition). These measures were selected as practical, clinically accessible indicators of nutritional status that could be uniformly applied across participants. We acknowledge the discrepancy between the written criterion and the measures used and appreciate the opportunity to address this.

9. Why is "bladder cancer" specifically listed as an exclusion criterion when "History of cancer within the last five years, other than local basal or squamous cell carcinoma," is already included?

There is an association between pioglitazone and an increased risk of bladder cancer, as highlighted by the FDA's recommendation to exercise caution when prescribing pioglitazone to individuals with a history of bladder cancer. To prioritize patient safety and adhere to these guidelines, we established a specific exclusion criterion to exclude individuals with any history of bladder cancer, regardless of the time since diagnosis. This approach was taken to minimize any potential risks associated with pioglitazone.

10. There are significant gender and racial differences between IBM muscle/sera and control muscle/sera. Could these differences have influenced the results? Why was matching not performed?

We appreciate the reviewer's important comment. The muscle samples used in our study were obtained from archived specimens in the Johns Hopkins Neuromuscular Biobank, which limited our ability to select perfectly matched controls. Due to the rarity of age-appropriate, histologically normal muscle biopsies, we prioritized control samples from older individuals to better approximate the age range of IBM patients. Unfortunately, due to limited availability, we were unable to match the control group to the IBM group by sex or race. We acknowledge that this could have influenced the metabolomic results and have now added this point explicitly to the limitations section of the manuscript.

11. Was any method of drug accountability used? This is not mentioned.

Thank you for pointing this out. All participants brought in their empty pill bottles to each study visit and were questioned about adherence. We added this to the Methods section under "Study Population and Design."

12. The manuscript states, "At the baseline visit, week 16, and week 32, a needle muscle biopsy from the vastus lateralis was performed." Were the biopsies taken from the same side and area of the vastus lateralis? Given that IBM is a patchy and asymmetrical disease, repeated biopsies may be problematic. If subsequent biopsies were taken from the same area, results could be affected by scar tissue. If taken from different locations, results could be influenced by varying levels of inflammation, degeneration, fat deposition, and mitochondrial changes, as these features are heterogeneously distributed in IBM muscle.

Thank you for providing us with the opportunity to clarify our methods. Needle muscle biopsies were performed on the same side and muscle for each patient to avoid variability given asymmetry of the disease. The site for each biopsy was localized first by muscle ultrasound to ensure viable muscle tissue and avoid scar or extensive fibrosis. This has been added to the Methods section.

13. Which methods and devices were used for assessing "knee extensor strength by quantitative dynamometry" and "hand-grip strength using handheld dynamometry"? Which side of the body was assessed? If both sides were assessed, was the average value used? This is not explained in the manuscript

A handheld dynamometer was used to assess quantitative muscle strength of the knee extensor. Participants were seated with their knee at 90 degrees of flexion. The dynamometer was placed on the distal part of the lower leg, just above the ankle, and manual resistance was applied gradually until the dynamometer recorded a measurement. Strength was measured in both lower extremities and then averaged.

Handgrip strength was measured using a Jamar hydraulic hand dynamometer following standardized procedures. Participants were seated with the elbow flexed at 90 degrees and the forearm in a neutral position. The test was performed on the standard (second) handle setting, and the best of three trials for each hand was recorded. The grip strength for each side was then averaged.

These details have been added to the Supplemental Methods. We also added more detailed information for the other clinical outcome measures as well.

14. Regarding the "modified timed up and go" (m-TUG) test, how did the authors handle patients who were unable to perform the test, mathematically/in the change analysis?

All participants were able to perform the m-TUG. Being able to rise from a chair was an inclusion criterion to ensure that participants would be able to complete this test.

15. The manuscript states, “Certified physical (AM, RP) and occupational therapists (MM)...” What does “certified” mean in the context of this trial? Which entity provided the certification? Did this include specific training on trial outcome assessments?

Thank you for the opportunity to clarify this point. The term "certified" refers to the fact that the physical (AM, RP) and occupational therapists (MM) involved in the trial hold professional licensure and are certified to practice in their respective fields. Their certification is granted by state licensure boards and professional organizations that govern physical and occupational therapy practice. While the therapists did not receive specific certification for clinical trial outcome assessments, we conducted a training session prior to the start of the trial to standardize the assessment process. This training ensured consistency in evaluating strength and applying assessment techniques, aligning with best practices for clinical care in IBM. This is now clarified in the manuscript.

16. The primary outcome is defined as “the change in PPARGC1A gene transcription measured by RNA sequencing after pioglitazone (week 32) compared to before treatment (week 16),” which corresponds to the intervention phase. However, the secondary outcome measures are described as changes “from baseline,” with baseline defined as week 0 (16 weeks before pioglitazone initiation, as per Figure 1). Why was a different time frame used for the secondary outcomes (week 0 to week 32) than for the primary outcome (week 16 to week 32)? This inconsistency is problematic.

Thank you for bringing to our attention this important inconsistency. We have made several modifications to the manuscript to ensure the time frame of comparison is consistent throughout primary and secondary outcomes and with the study protocol. One such critical adjustment was more closely aligning primary outcomes with those detailed in the study protocol and modifying the analysis of PPARGC1A expression to be a direct comparison of change from lead-in (baseline to week 16) to treatment phase (week 16 to 32).

17. The manuscript states, “Baseline demographic, clinical, and laboratory variables were evaluated with descriptive statistics, and differences were evaluated using the Wilcoxon rank sum test for continuous variables and Fisher’s exact test for categorical variables.” This sentence is unclear. What differences were being compared? It reads as though baseline values are being compared (e.g., between IBM and healthy subjects), in which case a paired test would not be appropriate.

The authors understand the sentence was likely confusing and lacked specificity. We have therefore updated it in the manuscript to clarify comparisons are being made between IBM and healthy subjects at baseline. The authors would also agree use of a paired test would not be appropriate, however in this case the Wilcoxon rank sum and Fisher’s exact tests used were not paired tests but rather standard independent statistical tests.

18. The manuscript states, “Genes with a ≥ 1.5 log₂ fold change in expression were considered to be significantly differentially expressed (at $P < 0.05$).” Given the large number of genes assessed, why was a more stringent p-value cut-off not applied? Why was a correction method such as Bonferroni not used?

The authors apologize for the inclusion of this sentence; it was unfortunately left in from a previous version of the manuscript which included an analysis of individual differentially expressed genes. The current manuscript utilizes Gene set enrichment analysis, a method in which cutoffs for individual gene significance are not used. We have therefore removed the sentence from the methods section as it is not applicable to the analyses performed in the study.

19. The manuscript states, “Epinephrine was detected in muscle samples.” Could this have influenced the expression of other metabolites?

Thank you for raising this important point. We acknowledge that the use of epinephrine can potentially impact the expression of other metabolites, however, we did not see any of these potential effects (ex. increased lactate due to vasoconstriction and anaerobic metabolism). We also note that the biopsies were quickly frozen, minimizing any potential for changes in more stable metabolites such as amino acids and fatty acids. Furthermore, because the aim was to assess longitudinal changes within the IBM group from pioglitazone, any potential influence of epinephrine would be consistent across all time points, minimizing its effect on the within-group comparisons.

Nevertheless, we recognize that the differences in biopsy procedures and anesthetic protocols between the IBM and healthy control groups may have contributed to differences in metabolite levels. We have included this as a limitation in the manuscript to provide appropriate context for interpreting our findings.

20. The analysis of changes in the IBM metabolic signature is difficult to interpret. How can a correlation between two pre-treatment values (baseline and week 16) be compared to a correlation between week 16 and week 32? How does this inform the treatment effect? This approach is unconventional. Why were week 16 and week 32 results not compared directly, as was done for the primary endpoint?

We appreciate the reviewers' concerns regarding the interpretations of the correlation analysis. As discussed further in reviewer comment 26 and our response, the lack of a placebo group acted as a limitation to our study design. To contextualize the effects observed during the treatment period (week 16 to week 32 comparison, figure 7B) on IBM metabolic signature we also included an analysis assessing the lead-in untreated phase (baseline to week 16 comparison, figure 7A). We believe this serves as a valuable “negative control” in the absence of a placebo group in that it critically demonstrates the observed effect (seen in treatment phase, 7B) was not present in the pre-treatment phase (7A) - providing some confidence the observed effect is not completely spurious. To the second point regarding the direct comparison of week 16 and week 32 for the primary endpoint, we have now updated this to be more in line with the original protocol and therefore more consistent with the correlation analyses focused on comparing lead-in phase to treatment phase.

21. Does “average duration of IBM” refer to the time since symptom onset or since diagnosis?

This refers to the time since symptom onset and has been clarified in the manuscript.

22. How was control muscle obtained? Was it taken from the vastus lateralis using the same technique as in IBM patients?

We obtained the control muscle samples from the Neuromuscular Biobank at Johns Hopkins University. They were collected from patients undergoing open surgical muscle biopsies for clinical indications, and we selected patients whose muscle symptoms were ultimately attributed to unrelated, non-myopathic conditions such as radiculopathy, and whose muscle biopsies were histologically normal. We prioritized older individuals to more closely approximate the age distribution of the IBM cohort. As muscle biopsies often come from varied muscle groups, some samples came from the rectus femoris, deltoid or biceps. While we acknowledge differences in fiber type composition across muscles, it is important to note that core metabolic pathways – such as glycolysis, the TCA cycle, and oxidative phosphorylation – are generally conserved across skeletal muscle, supporting the validity of our approach.

We recognize the differences in the biopsy procedure may introduce variability. However, the focus of our study was to identify an "IBM metabolic signature" and assess its longitudinal changes in response to pioglitazone, which were all conducted under consistent biopsy conditions for the IBM group.

We have added these details to the Methods section of the manuscript, and we acknowledge the different procedures used for controls in the limitations section. Thank you for your valuable feedback.

23. The manuscript states, "Untargeted metabolomics was performed on the 15 remaining samples and compared to five healthy control muscle specimens." These sample sizes are extremely low for metabolomics studies, raising concerns about the validity of the results and the potential influence of outliers. The same concern applies to the serum sample analyses.

We fully acknowledge that our sample size is small, which is a limitation of this study, and we now further emphasize this point in the discussion. However, we note that this is to our knowledge one of the largest metabolomics studies in IBM to date. To enhance the robustness of our findings, we not only identified an "IBM metabolic signature" through cross-sectional analysis (IBM vs. healthy controls) but also conducted a longitudinal analysis examining metabolic changes within the same individuals over time. By leveraging a within-subject design, we were able to reduce the impact of inter-individual variability and improve statistical power in detecting meaningful metabolic changes in response to pioglitazone. Had we relied solely on cross-sectional comparisons, we agree that the study would have been significantly underpowered.

We have clarified this in the manuscript to acknowledge the sample size limitation while highlighting the strengths of our longitudinal approach. We appreciate the reviewer's insightful feedback.

24. The manuscript states, "Patients with a US sum score ≥ 10 were characterized as having severe disease (n=7), and patients with a US sum score < 10 were characterized as having mild-moderate disease (n=6)." Has this cut-off ever been validated? What was the rationale for choosing it?

Thank you for this question. The rationale for this scoring system was the need to capture a muscle quality/structural assessment of the involved muscles in IBM as an objective measure of disease severity, augmenting the IBM-FRS which is subjective and can reflect functional compensation. Prior studies have shown that the most discriminating muscles are the FDP, rectus, and vastus lateralis which was why they were chosen, and this has been added in the manuscript. The degree of involvement was ascertained using the heckmatt 4 point scale (with 1 being normal and 4 having a marked increase in echointensity). Muscles

scored in the range of 3 and 4 represent more severe and obvious involvement, so a cutoff of > 10 as a composite score for the FDP, VL and rectus (with atrophy in any muscle) was used as this gave a clear distinction of a more advanced phenotype. While this US score is novel, we believe that a structural assessment of the muscle is very helpful to stratify disease and is currently being validated in a larger IBM study.

We caught an error in the manuscript which was corrected. In the methods, we indicated >10, but in results this was erroneously typed as ≥ 10 and has been amended.

25. The manuscript states, "Using a linear mixed-effects model, we observed increased PPARGC1A gene expression after pioglitazone, but this did not reach statistical significance ($p=0.07$)." Please clearly state that there was no treatment effect of pioglitazone on the primary endpoint rather than focusing on "trends." The same comment applies to the discussion section.

This result is no longer in the manuscript due to the adjustment in the analysis to better fit the pre-specified outcomes in the protocol. However, the Reviewer's point is well-taken and we have updated the language in the discussion so we do not over-interpret our findings.

26. The lack of a placebo group is a major limitation of this study. This should be further emphasized. Why was a placebo group not evaluated?

We agree that the lack of a placebo group is a major limitation of this study, and we have further emphasized this point in the discussion. A placebo-controlled trial in IBM would require a substantially larger sample size due to the considerable variability among IBM patients. Given that this was an exploratory study, our approach was to compare patients to themselves over time, which allowed us to detect metabolic changes with a much smaller sample size. This within-subject design helped mitigate inter-individual variability and improve statistical power despite the limited number of participants.

Importantly, our findings provide preliminary data that support the feasibility and rationale for a larger, placebo-controlled trial in IBM. We appreciate the reviewer's feedback and have clarified this in the manuscript.

27. Are the observed metabolomic changes specific to IBM? Would similar changes be expected in other conditions or even in healthy individuals receiving pioglitazone?

This is an important question that, to our knowledge, has not been directly investigated in healthy muscle. Prior studies have reported a similar favorable metabolic shift in muscle from individuals with insulin resistance, suggesting that some of these changes may not be exclusive to IBM. We have added several lines to the discussion acknowledging this knowledge gap and the need for further research.

28. The statistical methods section is unclear. Some statistical tests (e.g., Two-way ANOVA with Tukey HSD, mentioned in Figure 7) are referenced in figures but not described in the methods section.

The authors apologize for the lack of inclusion of details on the two-way ANOVA analysis in the methods sections. It has been updated to include these details such as the software and packages used to perform the test.

29. The manuscript states, “These metabolic responders tended to have earlier and less severe disease as stratified by US (Figure 4A, 4B, 7D).” However, these data are not convincing. Figure 7D shows only five patients in the mild-moderate group; if one “red dot” patient is removed, the effect would disappear, just as it is absent in the advanced group.

The authors understand the reviewers' concerns and agree the small sample size is a limitation to the study, which is likely made more concerning with subgroup analyses. To provide some measure of robustness we therefore performed a “leave-one-out” analysis. As the reviewer suspected, we did observe loss of significance in the severityXTreatment effect with removal of one of the 12 subjects (from $p=0.013$ to $p=0.103$). We have updated the manuscript to include mention of this additional analysis and further highlight limitations of the small sample size.

30. This is a negative study in IBM patients. Highlighting a single patient with subjective “clinical improvement” is inappropriate. For some outcome measures, the reported variation for this patient falls within the expected measurement error. Furthermore, given that this patient was obese and had metabolic syndrome, any observed “improvement” may simply reflect better cardiovascular and metabolic health rather than an IBM-specific effect.

We appreciate the reviewer’s feedback and agree that highlighting a single patient’s improvement is not appropriate in the context of a negative study. To address this concern, we have removed the description of this patient from the manuscript.

31. The manuscript states, “In a subset of patients, mostly with mild-moderate disease, a demonstrable metabolic response was seen in IBM muscle, which was associated with a slower decline in IBM-FRS and m-TUG scores.” Please tone down this statement, as the data are not convincing (see previous comments).

We appreciate this feedback and agree this statement should be toned down. It now reads:

“A subset of patients had a metabolic response to pioglitazone in muscle, and this metabolic response was associated with a slower decline in IBM-FRS and m-TUG scores.”

32. The statement, “These results provide initial evidence that pioglitazone can partially reverse the metabolic abnormalities of IBM muscle and that this reversal can translate into improved clinical outcomes for a subset of patients,” should be toned down, as the data do not convincingly support this conclusion.

We have toned down this statement as well. It now reads: “These results provide initial evidence that pioglitazone can partially reverse the metabolic abnormalities in IBM muscle, but larger trials are still needed to determine if this translates into improved clinical outcomes for patients.”

33. Table S2 states that comparisons between baseline values of IBM patients and controls were made using the paired Wilcoxon test. This is inappropriate because these are independent samples.

The authors agree use of a paired test would be inappropriate for comparison of the IBM patients to controls, however the data in table S2 was analyzed using a Wilcoxon rank-sum test and Fischer's exact test, neither of which was performed with a paired design.

34. Table S7 should be included in the main manuscript, as it clearly shows that the study was negative for all assessed physical function, strength outcome measures, and CK. For each outcome measure, please present their units, if applicable.

We agree with the reviewer and have incorporated Table S7 into the main manuscript as Table 1. We have included units for each outcome measure.

Reviewer #2 (Remarks to the Author):

This study is an exploratory, single-arm clinical trial investigating the effects of pioglitazone in patients with inclusion body myositis (IBM), with metabolomic analysis of biopsied muscle as the primary endpoint. Additionally, the study compares the metabolomic and transcriptomic profiles of patients at baseline with those of healthy controls from a biobank, allowing for a clearer interpretation of treatment-induced changes in the context of disease-specific alterations.

Despite the small sample size, the study is well-designed, incorporating a lead-in period during which needle muscle biopsies were performed. This design adds significant value beyond a simple pre- and post-intervention metabolomic comparison. The manuscript is well-written and provides valuable insights. Although the effects of pioglitazone were not clearly detected—partly due to the exploratory nature of the trial—the findings contribute meaningfully to the design of future clinical trials for this debilitating disease. My specific comments are as follows:

Major Points

1. Figure 4 effectively illustrates the relationship between ultrasound (US) severity and muscle metabolome. The authors may also wish to examine the associations between US findings and IBM-FRS, as well as between metabolome profiles and IBM-FRS.

The authors appreciate the reviewers' suggestions. To assess the association between US findings and IBM-FRS we performed a correlation analysis between US scores and IBM-FRS at baseline revealing a significant negative correlation, suggesting more severe US scores (higher value) were correlated with decreased function assessed by IBM-FRS (lower values) with a Spearman's rho = -0.62 at $p = 0.03$. We believe this provides valuable additional support for the manuscript, in demonstrating the relationship between US severity and clinical function and have added a sentence to the manuscript describing this relationship. With regards to assessment of associations between metabolome profiles and IBM-FRS, the authors believe the analysis as presented in figure 8 A and B demonstrates the relationship between changes in metabolome signature and IBM-FRS.

2. The study identifies a subset of responders based on muscle metabolic changes following pioglitazone treatment (Figure 7). To further elucidate the clinical significance of these metabolic alterations, the authors should compare motor performance and biomarker changes between responders and non-responders.

The authors agree with the reviewer that assessing motor performance and biomarker changes in responders vs non-responders would be of great interest. Unfortunately, due to the small sample size, such analyses are greatly limited in power and reliability. In particular, based on the observed change in “ibm metabolic signature” (Figure 7C) there are 4 subjects who we would consider responders (significant difference in lead-in vs treatment phase, based on an additional analysis comparing jackknife estimates of CI and only considering those with no overlap between phases). We therefore believe our current method of evaluating the clinical relevance of the metabolic score itself provides a slightly larger sample size from which to determine if metabolic response is likely to correspond to clinical outcomes (figure 8). We have, however, performed an additional more direct analysis as the reviewer suggested modifying our original linear-mixed effects models for clinical outcomes to now include an additional “responder” variable, and report in the table below the model estimates for timeXTreatment[Pio]Xresponder[Responder]. While largely demonstrating null findings, the results do indicate that the small subset of responders showed a significant improvement in IBM-FRS over time with treatment compared to non-responders who saw decreases.

Outcome	estimate	p-value
IBM-FRS	5	0.002
Kendall Score	0.19	0.9
Fi2	-0.07	0.52
Hand-grip average	5.56	0.091
Creatine kinase	-283.61	0.303
6-minute walk	-24.64	0.86
Dynaamometry knee extensors	10.27	0.169
M_TUG	0.47	0.704

3. To strengthen the multi-omics approach, the authors should directly compare treatment-induced transcriptomic changes (Figure 6) with metabolomic alterations (Figure 7).

The authors thank the reviewer for the suggestion. As suggested, we have performed an additional analysis integrating gene set enrichment analysis results across both omics layers to reinforce our findings using the multiGSEA package. These results are mentioned in the updated manuscript and the output is provided as a supplementary table (Table S8).

4. Similarly, a comparison between disease-related transcriptomic changes and treatment-induced transcriptomic alterations would provide additional valuable insights.

The authors agree that a comparison between disease-related transcriptomic changes and treatment-induced transcriptomic changes would be informative. Unfortunately, transcriptomic data from our

control samples is not available as budget constraints did not allow for these analyses at the time. To gain some insight into this question, however, the authors have utilized a publicly available transcriptomic dataset (GSE102138) containing control and IBM skeletal muscle samples to act as a “transcriptomic disease signature”. Intriguingly, the resulting correlations between lead in and treatment period with this disease signature align with our metabolomic signature results and even suggest potentially increased sensitivity of the transcriptome to detect disease progression in the lead in phase. We report a Spearman's rho of lead-in vs IBM signature of 0.4 with $p=0$, indicating a more “IBM-like” transcriptome profile over the lead-in period, while there was a Spearman's rho of -0.54 at $p=0$ during the treatment period suggesting a moderate reversal of transcriptome signature. We believe these results provide valuable support for our findings in the metabolome by demonstrating a similar pattern at another “omics layer” and providing some indication of external validity/generalizability for IBM disease signatures. The authors thank the reviewer for the suggestion and have added these results as a supplemental figure (Figure S1).

Minor Points

1. Was the muscle biopsy site in the control group the same as in the patient group (i.e., vastus lateralis)?

Thank you for the question. As we addressed in reviewer #1's question: We obtained these samples from the Neuromuscular Biobank at Johns Hopkins University. They were collected from patients undergoing open surgical muscle biopsies for clinical evaluation, and we selected patients whose muscle symptoms were ultimately attributed to unrelated conditions such as radiculopathy, and whose muscle biopsies were histologically normal. We prioritized older individuals to more closely match the IBM cohort. As muscle biopsies often come from varied muscle groups, samples came from the rectus femoris, deltoid or biceps. While we acknowledge differences in fiber type composition across muscles, it is important to note that core metabolic pathways – such as glycolysis, the TCA cycle, and oxidative phosphorylation – are generally conserved across skeletal muscle, supporting the validity of our approach. However, we have included this as a limitation in the discussion.

2. A more detailed discussion is needed regarding the possible causes of heart failure exacerbation in one patient.

Thank you for this comment. Pioglitazone can cause or exacerbate CHF in some patients and is contraindicated in those with class III or IV heart failure. This patient did not have any clinical signs or symptoms of heart failure at the time of enrollment but had this in their remote history as occurring after

a virus, which then resolved. Two weeks prior to termination of the trial the patient felt a sense of chest discomfort which gradually worsened with finding of left sided pleural effusion. Pioglitazone was stopped and diuretics were given with resolution of symptoms. More detail on this patient was included in the paper.

3. There is an ethical concern regarding the re-initiation of pioglitazone in one patient. How did the ethical review committee justify this treatment outside of the study protocol?

We appreciate the reviewer's concern regarding the ethical considerations of pioglitazone re-initiation in one patient. In response to this and related feedback, we have removed the discussion of this individual's post-trial pioglitazone use and subsequent treatment course from the manuscript.

To clarify, the patient was treated with pioglitazone strictly according to the approved study protocol during the trial period. After completion of the trial and discontinuation of study treatment, the patient and their independent treating physician—outside the direction or involvement of the study team—elected to re-initiate pioglitazone use as an off-label clinical decision. This occurred outside the scope of the study and was not subject to oversight by the study's ethics committee. The study team became aware of this off-protocol treatment retrospectively, when the patient shared this information during participation in a separate, observational longitudinal study of IBM. We agree that inclusion of this anecdotal clinical case may have introduced confusion regarding the scope and oversight of the clinical trial, and we have therefore removed it from the revised manuscript

Reviewer #3 (Remarks to the Author):

Reviewer #5 (Remarks to the Author):

General Comments

This was a small study which aimed to evaluate the impacts of pioglitazone on various outcomes and assessed the differences between healthy muscle specimens and serum samples with those from IBM patients. PPARGC1A expression increased at week 32 compared to week 16 (before treatment) $P=0.07$, indicating that pharmacologically targeting mitochondria with pioglitazone may improve metabolic deficits in this population.

For a single arm study, the "lead-in" phase was a helpful addition to assess the stability of the clinical outcomes assessed. The PPARGC1A gene expression (primary outcome) increased compared to pre-treatment timepoints however it is difficult to assess the extent of this since little information is reported about this outcome's results.

Small sample size and no controls does limit the study. More patients would have given the authors a better understanding of the effect pioglitazone. The lack of controls does mean that it is hard to assess a potential Hawthorne effect, however due to primarily biological outcomes (which can still be affected by being in a

trial), and nature of the small/exploratory study design I don't think this is a major flaw with the trial.

It is unclear if the statistical analyses were pre-specified prior to screening.

I agree with the authors' conclusions.

Main Comments

“Control muscle specimens were obtained from the Johns Hopkins Neuromuscular Biobank and were collected from patients with normal muscle tissue and no diagnosed neuromuscular disorders.” and “Serum samples were obtained from healthy controls as part of a separate IRB-approved study for healthy donors.”

It is not clear what defines “normal muscle tissue or a healthy control?” Were these just samples from individuals without Inclusion body myositis or without a set of conditions? Were they randomly selected from the biobank/ healthy donors in the IRB-approved study or were they all samples the bank/study had at the time? Without this detail, it would be hard to reproduce the work.

Thank you for this comment. We agree that more information on the controls is needed to improve the reproducibility of this study.

The control muscle samples were obtained from archived specimens in the Johns Hopkins Neuromuscular Biobank. They were collected from patients undergoing open surgical muscle biopsies for clinical indications, and we selected specimens from patients whose muscle biopsies were reported as normal, and whose symptoms were ultimately attributed to unrelated, non-myopathic conditions (such as radiculopathy). Any sample where a primary muscle disease (such as mitochondrial myopathy) could not be ruled out was excluded. We also prioritized biopsies from older individuals to more closely approximate the age distribution of the IBM cohort. This information has been added to the Methods section of the manuscript.

Healthy control serum came from a separate protocol which obtained blood samples from healthy donors. Exclusion criteria for this cohort included pregnancy or a history of autoimmune disease, cancer, or active tuberculosis, HIV, or hepatitis infection. This also has been added to the manuscript.

“Using a linear mixed effects model, we observed increased PPARGC1A gene expression after pioglitazone, but this did not reach statistical significance ($p=0.07$).”

This is missing an effect size estimate and confidence interval. Since this is the primary outcome, I would have expected this to be clearly written. I think the interpretation of the P-value could be potentially less conservative with their (particularly considering the small sample size). They did observe that there was an increased PPARGC1A gene expression after pioglitazone. The plot for this (Figure 6A) also hides the confidence interval for the PPARGC1A Normalized Averages of Counts due to the black colour for the bars and the error bars.

The authors appreciate the suggestions. We have revised the description and presentation of the primary outcome to more closely align with the original study protocol. We now include the effect size and confidence interval for the change in PPARGC1A gene expression. In addition, we have updated the figure to improve visualization.

Minor comments

4. "Model equations were "outcome ~ 1+Treatment*Time + (1+Time|ID)"

Did the authors consider adjusting for the baseline demographics they collected (age, sex, ethnicity, disease duration, biomarkers etc.)?

The reviewer raises an important point regarding model adjustment. While ideally several demographic factors could be adjusted for, given the extremely small cohort size such adjustments can be limited. In particular, some analyses were attempted with inclusion of sex as a covariate and it was found that this generally produced suspected overfitting evidenced by higher AIC and BIC scores compared to unadjusted models. It was therefore decided to use unadjusted models. We do, however, agree this is a point worth raising and have added a sentence discussing it in our limitations.

5. "The mean age of control muscle specimens was 60.6 ± 15.4 years and 4/5 (80%) were female (n=5). The control sera was slightly younger than the IBM sera (56.3 ± 5.7 years, $p=0.001$) and 7/10 (70%) were female (n=10)."

The "(n=5)" and "(n=10)" at the end of these sentences is slightly confusing as it is not clearly referring to something. I presume this is simply referring to the number of control sera and muscle specimens but this could be clearer.

We have made this clearer. It now reads as follows: "The mean age of the 5 control muscle specimens was 60.6 ± 15.4 years and 4/5 (80%) were female. The 10 control sera were slightly younger than the IBM sera (56.3 ± 5.7 years, $p=0.001$) and 7/10 (70%) were female."

6. "Using principal component analysis, we found that samples from healthy control and IBM muscle showed distinct clustering patterns, with a more variable clustering of IBM samples than healthy controls (Figure 2A)."

Though this is only 5 healthy controls so could be due to randomness/only having 5 controls.

The authors understand the low sample size may cause concern. To more precisely describe potential differences in variance between groups we therefore conducted an F-test on PC1 values for controls vs IBM to assess more robustly if there are statistically significant differences in variance. The results indicate a statistically significant difference in variability between the two groups with $p = 0.018$, providing some support to the robustness of these observations. We have added this to the manuscript.

7. "This stratification correlated with function and those with severe disease by US had lower IBM-FRS scores (28.57 ± 3.73 vs 33.33 ± 3.72)."

Are these recorded anywhere in a table/figure? I can't see them reported elsewhere.

We have included a table of the ultrasound score and IBM-FRS for each patient at the baseline visit in supplemental Table S5.

Patient	US score	Atrophy present	IBM-FRS
1	11	Yes	32

2	5	No	35
5	12	Yes	28
7	11	Yes	22
8	12	Yes	26
10	12	Yes	28
11	9	No	30
12	12	Yes	30
13	10	No	32
14	7	No	39
16	11	Yes	33
17	8	No	29
18	8	No	32

8. “We did not observe any difference in the metabolic signature of severe versus mild-moderate IBM in sera (Spearman’s $p = -0.13$, $p = 0.06$) (Figure 4D).”

It is stated that the investigators did not observe “any” difference however there was a small difference observed. Whether this was due to chance or if there is a true difference is a slightly different matter.

We have updated the language in the manuscript to be more precise, it now reads “...a significant difference...”.

9. “Despite notable agreement for a few key metabolites, such as propionylcarnitine and tryptophan which were both decreased in serum and muscle, the overall correlation between the metabolic profiles in muscle and serum was not significant (Figure 5E, Spearman’s $p = 0.06$).”

As with the statement regarding correlation between fold change between IBM vs. controls and severe vs mild IBM (and various other statements), it would be good for the P-value to be stated here also ($P = 0.501$)

We have added the p-value to this segment of the manuscript.

10. “After completing the clinical trial, the patient discontinued pioglitazone and his CK increased back to his pre-treatment baseline of 2055 units/L,”

CK units have been referred to as U/L throughout apart from this sentence.

Thank you for catching this error. We have deleted this paragraph per the suggestion of Reviewers 1 and 2.

11. “Although there was no statistically significant change in disease trajectory for the study population as a whole, one patient (subject 11) demonstrated a clinical improvement with pioglitazone.”

Some context around why this participant was followed up so closely after the end of the trial might be helpful since I can’t see this mentioned in the methods (was this planned or just because of their promising

CK levels during the trial). Were any other patients followed-up post 48 weeks? By chance you would expect someone to have a strong reduction in levels of one of these 7 outcome measures so inference from this patient's data should be limited (as I think it fairly is).

To clarify, all participants were prospectively followed only through the 48-week study period, with no planned protocolized follow-up beyond that timepoint. Subject 11 was concurrently enrolled in a separate longitudinal study of inclusion body myositis (IBM) being conducted at our institution. As part of that separate study, additional clinical data were collected after completion of the trial. We agree with the reviewer that inference from a single patient's post-trial clinical course should be limited, particularly because such post-treatment information was not prospectively collected as part of the present study. In response to this and other reviewer feedback, we have removed detailed discussion of this patient's post-trial course to avoid potential overinterpretation or confusion regarding the scope of the trial and its prespecified outcomes.

12. "despite missing the primary endpoint of increasing PPAR γ gene expression"

The choice of words here is slightly misleading/confusing – the primary endpoint was not missed. There was not as strong evidence as was hoped for but there was still some evidence. *We appreciate the feedback. Discussion of the primary endpoint has been updated to more closely align it with the study protocol.*

We thank the Editor and Reviewers for their thoughtful review of our manuscript, which has helped to further improve its quality. We have addressed the two minor comments from Reviewer #1 and Reviewer #5, as detailed below, and have made the corresponding changes to the manuscript. In addition, we have added the DOI for the data and code, which are stored in the Vivli repository.

Reviewer #1 (Remarks to the Author):

One minor comment: The manuscript now states that “Eligible subjects included adults 50 years and older (to enrich for those with typical IBM features) with a diagnosis of clinico-pathologically defined or clinically defined IBM by ENMC criteria (18) or data-driven criteria (DDC) (19)”. However, in the protocol only ENMC criteria are mentioned. This discrepancy still requires clarification.

We apologize for this discrepancy. All of the participants met ENMC criteria. Some participants also met the DDC criteria, but this additional criteria is not necessary and not required by the study protocol so we have removed it from the manuscript.

Reviewer #5 (Remarks to the Author):

Thank you for your active responses to these comments.

Point 8 was:

8. “We did not observe any difference in the metabolic signature of severe versus mild-moderate IBM in sera (Spearman’s $p = -0.13$, $p = 0.06$) (Figure 4D).”

It is stated that the investigators did not observe “any” difference however there was a small difference observed. Whether this was due to chance or if there is a true difference is a slightly different matter.

Author response:

We have updated the language in the manuscript to be more precise, it now reads “...a significant difference...”.

I appreciate the efforts to correct this however, it would be inappropriate to say a significant difference. Instead a better interpretation is "some difference..."

We apologize for being unclear with the changes made to the manuscript. The current sentence reads, “We did not observe a significant difference in the metabolic signature of severe versus mild-moderate IBM in sera.” To clarify that this was not significant, we added the word “statistically.” The sentence now reads: “We did not observe a statistically significant difference in the metabolic signature of severe versus mild-moderate IBM in sera.”